# A crowd of BashTheBug volunteers reproducibly and accurately measure the minimum inhibitory concentrations of 13 antitubercular drugs from photographs of 96-well broth microdilution plates

**Philip W Fowler[1]\*, Carla Wright[1], Helen Spiers[2,3], Tingting Zhu[4], Elisabeth ML Baeten[5], Sarah W Hoosdally[1], Ana L Gibertoni Cruz[1], Aysha Roohi[1], Samaneh Kouchaki[4], Timothy M Walker[1], Timothy EA Peto[1], Grant Miller[2], Chris Lintott[2], David Clifton[4], Derrick W Crook[1], A Sarah Walker[1], The Zooniverse Volunteer Community, The CRyPTIC Consortium**

[1]Nuffield Department of Medicine, University of Oxford, Oxford, United Kingdom; [2]Zooniverse, Department of Physics, University of Oxford, Oxford, United Kingdom; [3]Electron Microscopy Science Technology Platform, The Francis Crick Institute, London, United Kingdom; [4]Institute of Biomedical Engineering, University of Oxford, Oxford, United Kingdom; [5]Citizen Scientist, c/o Zooniverse, Department of Physics, University of Oxford, Oxford, United Kingdom

**\*For correspondence:**
philip.fowler@ndm.ox.ac.uk

**Group author details:**
The CRyPTIC Consortium See page 17

**Abstract** Tuberculosis is a respiratory disease that is treatable with antibiotics. An increasing prevalence of resistance means that to ensure a good treatment outcome it is desirable to test the susceptibility of each infection to different antibiotics. Conventionally, this is done by culturing a clinical sample and then exposing aliquots to a panel of antibiotics, each being present at a pre-determined concentration, thereby determining if the sample isresistant or susceptible to each sample. The minimum inhibitory concentration (MIC) of a drug is the lowestconcentration that inhibits growth and is a more useful quantity but requires each sample to be tested at a range ofconcentrations for each drug. Using 96-well broth micro dilution plates with each well containing a lyophilised pre-determined amount of an antibiotic is a convenient and cost-effective way to measure the MICs of several drugs at once for a clinical sample. Although accurate, this is still an expensive and slow process that requires highly-skilled and experienced laboratory scientists. Here we show that, through the BashTheBug project hosted on the Zooniverse citizen science platform, a crowd of volunteers can reproducibly and accurately determine the MICs for 13 drugs and that simply taking the median or mode of 11–17 independent classifications is sufficient. There is there-fore a potential role for crowds to support (but not supplant) the role of experts in antibiotic suscep-tibility testing.

## Editor's evaluation

The authors evaluate a novel crowd–sourcing method to interpret minimum inhibitory concentrations of *Mycobacterium tuberculosis*, the causative agent of tuberculosis. To provide valuable test results without the need for available expert mycobacteriologists, the authors demonstrate that when presented appropriately, 11–17 interpretations by lay interpreters can provide reproducible results for most tuberculosis drugs. This analysis demonstrates that among those samples that can

be reliably interpreted by automated detection software, lay interpretation provides a potential alternative means to provide a timely confirmatory read. The work will be of interest to bacteriologists and those with an interest in antimicrobial resistance.

## Introduction

Tuberculosis (TB) is a treatable (primarily) respiratory disease that caused illness in ten million people in 2019, with 1.4million deaths (*World Health Organization, 2020*). Ordinarily this is more than any other single pathogen, although SARS-CoV-2 killed more people than TB in 2020 and is likely to do so again in 2021. Like most bacterial diseases treated with antibiotics, an increasing proportion of TB cases are resistant to one or more drugs.

Tackling this 'silent pandemic' will require action on several fronts, including the development of new antibiotics, better stewardship of existing antibiotics and much wider use of antibiotic susceptibility testing (AST) to guide prescribing decisions (*O'Neill, 2016*). The prevailing AST paradigm is *culture-based*: a sample taken from the patient is grown and the pathogen identified. If required, further samples are cultured in the presence of different antibiotics and each test is inspected/measured to see which compounds inhibit the growth of the bacterium. A scientific laboratory, with an enhanced biosafety level and staffed by experienced and highly trained laboratory scientists, is required to carry out such AST. Compared to most pathogenic bacteria, *M. tuberculosis* is unusually difficult,mainly because its doubling time is so slow resulting in culture times being weeks rather than days. Maintaining such laboratories with a cadre of expert scientists is expensive and hence they tend to be found only at larger hospitals and national public health agencies, even in high-income countries. This model, whilst effective, is practically and economically difficult to scale up which explains in part why most antibiotic prescribing decisions are still done without any AST data.

The *minimum inhibitory concentration* (MIC) of an antibiotic is the lowest concentration that prevents growth of the pathogen. Mycobacterial samples are usually inoculated onto an appropriate growth medium that contains an antibiotic at a single critical concentration; if the MIC of the same is above the critical concentration it will grow and hence be labelled resistant, otherwise there is no growth and it will be classified as susceptible. Although simple, the accuracy of such an approach decreases when the expected MIC is of the same order as the critical concentration, as is the case for many antituberculars (*CRyPTIC Consortium, 2022*). Historically, it has been assumed that, since accuracy is paramount, only highly-trained and experienced laboratory scientists (experts), or more recently, extensively-validated automatic algorithms part of accredited AST devices, can assess if a pathogen is growing and hence determine whether the sample is resistant or not.

In this paper, we shall show that a crowd of volunteers, who have no microbiological training, can reproducibly and accurately determine the growth of *M. tuberculosis* (the causative agent of TB) on a 96-well plate and thence the MICs for 13 different antibiotics. This is an application of the "wisdom of crowds" which can be traced back to observations made by *Galton, 1907*. An even earlier use is the Germanic definition of the foot as one sixteenth the combined length of the left feet of the first sixteen men to leave a church on a Sunday (*Koebel, 1563*). The main goal of the BashTheBug citizen science project, which was launched in April 2017 on the Zooniverse platform, is to reduce MIC measurement error in the large dataset of 20,637 clinical *M. tuberculosis* isolates collected by the Comprehensive Resistance Prediction for Tuberculosis: an International Consortium (CRyPTIC) project.

## Results
### Project launch and progress

After a successful beta-test on 22 March 2017, BashTheBug was launched as a project on the Zooniverse citizen science platform on 8 April 2017. The launch was publicised on social media, mainly Twitter, and mentioned on several websites and the Zooniverse users were notified via email. By the end of the first week, 2029 people had participated (of which 1,259 had usernames) classifying a total of 74,949 images – this includes the beta-test. The initial set of images were completed on 11 October 2017 and a second set was classified between 8 June 2020 and 15 November 2020 (*Figure 1A*) to increase the number of classifications per drug image from 11 to 17.

**eLife digest** Tuberculosis is a bacterial respiratory infection that kills about 1.4 million people worldwide each year. While antibiotics can cure the condition, the bacterium responsible for this disease, *Mycobacterium tuberculosis,* is developing resistance to these treatments. Choosing which antibiotics to use to treat the infection more carefully may help to combat the growing threat of drug-resistant bacteria.

One way to find the best choice is to test how an antibiotic affects the growth of *M. tuberculosis* in the laboratory. To speed up this process, laboratories test multiple drugs simultaneously. They do this by growing bacteria on plates with 96 wells and injecting individual antibiotics in to each well at different concentrations.

The Comprehensive Resistance Prediction for Tuberculosis (CRyPTIC) consortium has used this approach to collect and analyse bacteria from over 20,000 tuberculosis patients. An image of the 96-well plate is then captured and the level of bacterial growth in each well is assessed by laboratory scientists. But this work is difficult, time-consuming, and subjective, even for tuberculosis experts. Here, Fowler et al. show that enlisting citizen scientists may help speed up this process and reduce errors that arise from analysing such a large dataset.

In April 2017, Fowler et al. launched the project 'BashTheBug' on the Zooniverse citizen science platform where anyone can access and analyse the images from the CRyPTIC consortium. They found that a crowd of inexperienced volunteers were able to consistently and accurately measure the concentration of antibiotics necessary to inhibit the growth of *M. tuberculosis.* If the concentration is above a pre-defined threshold, the bacteria are considered to be resistant to the treatment. A consensus result could be reached by calculating the median value of the classifications provided by as few as 17 different BashTheBug participants.

The work of BashTheBug volunteers has reduced errors in the CRyPTIC project data, which has been used for several other studies. For instance, the World Health Organization (WHO) has also used the data to create a catalogue of genetic mutations associated with antibiotics resistance in *M. tuberculosis*. Enlisting citizen scientists has accelerated research on tuberculosis and may help with other pressing public health concerns.

## Volunteers

In total, 9372 volunteers participated in classifying this dataset doing a total of 778,202 classifications (*Figure 1—figure supplement 1*). The number of citizen scientists is an over-estimate since users who did not register with the Zooniverse (and therefore could not be identified through their unique username) but did more than one session will be counted multiple times. On 251 of the 356 days (70 %) when these data were being collected there were ten or more volunteers active on the project (*Figure 1B*), and on 31% of the days 50 or more volunteers were active. This illustrates the importance of engaging with the volunteers via multiple channels to ensure an active and sustainable project. Overall, this is a mean of 84.1 classifications per volunteer; however, this hides a large amount of variation in the number of classifications done by individual volunteers. Almost half of the volunteers (4299) did ten or fewer classifications and 1115 classified only a single image whilst the ten volunteers who participated the most did 103,569 classifications between them which is 13.1% of the total. The Gini-coefficient is a useful way to measure these unequal levels of participation, and for this dataset it is 0.85 (*Figure 1C*).

## Comparison to other Zooniverse projects

The activity within the first 100 days of launch has been used to benchmark and compare different Zooniverse projects from several academic disciplines (*Spiers et al., 2020*). A total of 381,964 classifications were done in the first hundred days after launch by a total of 6237 users of which 3733 were registered and so were unique. Several Zooniverse projects have attracted many more users and classifications; however, these are all ecology or astronomy projects which are the mainstay of the Zooniverse.

Since the number of classifications is heavily influenced by the difficulty of the task, it can be more illuminating to compare the Gini coefficients of different projects. *Cox et al., 2015* measured

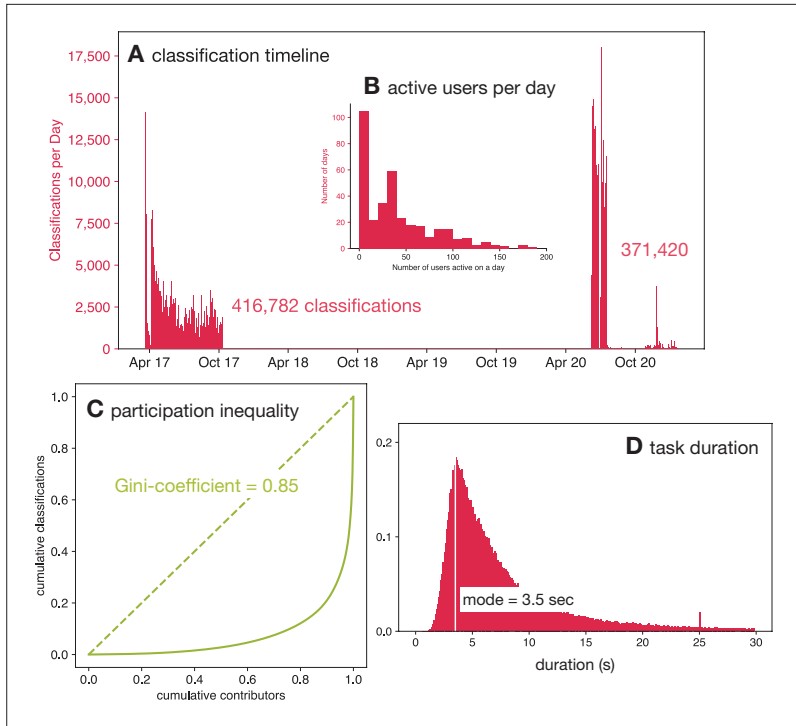

**Figure 1.** This dataset of 778,202 classifications was collected in two batches between April 2017 and Sep 2020 by 9372 volunteers. (**A**) The classifications were done by the volunteers in two distinct batches; one during 2017 and a later one in 2020. Note that the higher participation during 2020 was due to the national restrictions imposed due to the SARS-Cov-2 pandemic. (**B**) The number of active users per day varied from zero to over 150. (C) The Lorenz curve demonstrates that there is considerable participation inequality in the project resulting in a Gini-coefficient of 0.85. (D) Volunteers spent different lengths of time classifying drug images after 14 days of incubation with a mode duration of 3.5 s.

The online version of this article includes the following figure supplement(s) for figure 1:

**Figure supplement 1.** Thank you to all the volunteers who contributed one or more classifications to this manuscript.

**Figure supplement 2.** The time spent by volunteers on each classification varied with a mode of 3.5 s.

**Figure supplement 3.** The time spent by volunteers on each classification varied depending on the drug being considered.

**Figure supplement 4.** Every new user is shown this tutorial when they first join the BashTheBug Zooniverse project.

a mean Gini coefficient across several Zooniverse projects of 0.81, whilst a later and more comprehensive study *Spiers et al., 2019* demonstrated that Zooniverse projects had Gini coefficients in the range 0.54–0.94 with a mean of 0.80. They also suggested that biomedical projects had lower Gini coefficients, with a mean Gini coefficient of 0.67; however, this was only based on three projects. BashTheBug attracted more users, completed more classifications and had a higher Gini coefficient than any of these three biomedical projects (*Spiers et al., 2019*). A more recent biomedical project, Etch-a-cell, that launched at a similar time to BashTheBug had a Gini coefficient of 0.83 (*Spiers et al., 2020*). BashTheBug therefore has a higher than average level of participation inequality, having the 17th highest Gini coefficient out of 63 Zooniverse projects surveyed (*Spiers et al., 2019*).

## Time spent

The time spent by a volunteer classifying a single drug image varied from a few seconds up to hours; the latter are assumed to be data artefacts, for example when a volunteer leaves a browser window open. The distribution of time spent per image shows no appreciable differences when calculated as function of the incubation time with a mode of 3.5 s (*Figure 1D*, *Figure 1—figure supplement 2*),

which is unexpected given after only 7 days of incubation there is little or no bacterial growth. After 14 days incubation there are, however, observable differences between how long the volunteers spent classifying each drug (*Figure 1—figure supplement 3*).

## Classification validity

The tutorial on the Zooniverse website (*Figure 1—figure supplement 4*) encouraged volunteers to check that the control wells both contain bacterial growth – if not then the drug image should be marked as having 'No Growth in either of the 'No Antibiotic' wells'. They were also asked to check if any of the drug wells contain growth very different to all the others (contamination), inconsistent growth (skip wells), or anything else that would prevent a measurement being taken (artefacts). If any of these are true, they were asked to mark the drug image as "Cannot classify". In the analysis these were aggregated into a single dilution (NR – not read). In all cases, if a simple majority make a classification of NR, then this is always returned as the result. All NR results are excluded from calculations of the exact or essential agreement.

## Expert measurements

Each drug image was also measured by a laboratory scientist using a Vizion instrument as well as programmatically by some software, AMyGDA (*Fowler, 2017*). Although the AMyGDA software is reproducible, it will often classify artefacts, such as air bubbles, contamination, shadows and sediment, as bacterial growth and is also likely to assess a well as containing no bacterial growth when the level of growth is very low. By contrast, laboratory scientists are less consistent but can recognise and ignore artefacts. Since the sources of error for each these methods are different, we constructed a consensus dataset with an assumed reduced error rate by only including readings where both of these independent methods agree on the value of the MIC. We will refer to this as the 'Expert +AMyGDA' dataset and the larger dataset simply consisting of all the readings taken by the laboratory scientist as the 'Expert' dataset. We shall further assume the error-rate in the 'Expert +AMyGDA' consensus dataset is negligible, allowing us to use it as a reference dataset, which in turn will allow us to infer the performance of the volunteers by comparison.

A total of 12,488 drug images were read after 14 days incubation (*Supplementary file 1*); for 6205 (49.7%) of these both the laboratory scientist (Expert) and the software (AMyGDA) returned the same MIC. Since a laboratory scientist would be reasonably expected to make an error $\leq$ 5% of the time, the majority of the drug images excluded are most likely due to AMyGDA incorrectly reading a drug image with only a minority being genuine errors.

By constructing the consensus Expert+AMyGDA dataset we are likely to have introduced bias by unwittingly selecting for samples which are easier to read. One possibility is that we may have selected samples with higher than average levels of bacterial growth. We can show that this is not the case since not only is the average level of growth in the positive controls (as measured by AMyGDA) for the Expert+AMyGDA dataset (30.4%) similar to that observed (30.6%) for the larger Expert dataset (*Figure 6—figure supplement 3*), but the distributions themselves are very similar.

A second possibility is that drug images with specific growth configurations (for example either no growth or growth in all the wells) are easier to read than drug images where the growth halts. This would imply that the probability of the Expert+AMyGDA measurements agreeing is a function of the dilution MIC, which indeed is what we find (*Supplementary file 1d*). The agreement is highest when there is no growth in any of the drug wells, which makes sense as that is a relatively trivial classification to make. The next highest value is when the dilution is 8, which since 7 of the 14 drugs on the plate have 7 drug wells (Figure 6), corresponds to growth in all 7 drug wells, which is also an easy classification.

The net effect of this is that the Expert+AMyGDA dataset has a different distribution of measured MICs, including a greater proportion of drug images with a low MIC dilution (61.4% after 14 days incubation have a dilution of 1 or 2, *Figure 6—figure supplement 4*) compared to the parent Expert dataset (45.8%). One should bear in mind this bias when interpreting the results, and to assist we will consider if key results change when we use the Expert-only dataset.

## How to compare?

Ideally one would apply an international standard for antibiotic susceptibility testing (AST) for Mycobacteria which would permit us to assess if a consensus measurement obtained from a crowd of

volunteers is sufficiently reproducible and accurate to be accredited as an AST device. Unfortunately, there is no international standard for Mycobacterial AST – the need to subject Mycobacteria to the same processes and standards as other bacteria has been argued elsewhere (*Schön et al., 2020*) – we shall therefore tentatively apply the international standard for aerobic bacteria (*International Standards Organization, 2007*) which requires the results of the proposed antibiotic susceptibility testing method to an appropriate reference method.

Neither of the measurement methods used in constructing our reference consensus dataset has been endorsed, although broth microdilution using Middlebrook 7H9 media was recently selected by EUCAST as a reference method for determining *M. tuberculosis* MICs (*Schön et al., 2020*) but only for manually-prepared 96-well plates, permitting much larger numbers of wells for each drug. Nor has any software-based approach for reading MICs from 96-well microdilution plates been endorsed by EUCAST, the CLSI or any other international body. Despite this, and in lieu of any other reasonable approach, we shall treat the consensus MICs (the Expert +AMyGDA datset) as a reference dataset and apply ISO 20776–2 (*International Standards Organization, 2007*).

This requires a new AST method that measures MICs to be compared to the reference method using two key metrics: the *exact agreement* and the *essential agreement*. The former is simply the proportion of definite readings which agree, whilst the latter allows for some variability and is defined as the "MIC result obtained by the AST device that is within plus or minus doubling dilution step from the MIC value established by the reference method" (*International Standards Organization, 2007*). To meet the standard any new AST method must be ≥95% reproducible and ≥90% accurate (both assessed using essential agreement) compared to the reference method (*International Standards Organization, 2007*).

## Variability in classifications

Inevitably there is a large degree of variation in the classifications made by different volunteers of the same drug image. Examining all 112,163 classifications made by the volunteers on the 6205 drug images taken after 14 days incubation and comparing them to the consensus of the laboratory scientist and AMyGDA shows that a single volunteer is likely to exactly agree with the Expert +AMyGDA dataset 74.6% of the time, excluding cases where either concluded the drug image could not be read (*Supplementary file 1c*). This rises to 86.4% when only considering essential agreement.

The magnitude of agreement varies depending on the measured dilution: if the consensus view is that a drug image contains no growth, a single volunteer is likely to agree 64.1% of the time (*Figure 2A*), however this falls to 47.8% if the consensus dataset indicates that the first four wells contain growth before rising to 94.5% when the laboratory scientist decides the dilution is 8. We hence recapitulate our earlier observation that drug images with no growth or growth in all wells are easier to read than drug images where only a subset of drug wells contain growth.

The BashTheBug volunteers are likely to return a higher dilution than the Expert +AMyGDA consensus; this can be seen in the greater proportion of MICs with higher dilutions (*Figure 2B–D*). For example a single volunteer is at least 5 ×, and often > 10 ×, more likely to return an MIC one greater than the reference rather than an MIC one lower than the reference (*Figure 2D*). We shall return to this bias later.

When the classification made by an individual volunteer does not agree with the consensus this is often (but not always) because they have misclassified the drug image (*Figure 2E–G*). Comparing the classifications made by individual volunteers with the larger, but presumably less accurate, Expert dataset we see that an individual volunteer is less likely to agree with a single laboratory scientist with the overall level of exact agreement falling from 74.6% to 65.3% (*Supplementary file 1*, *Figure 2— figure supplement 1*). Regardless of the comparison dataset used, it is clear that to achieve satisfactory levels of reproducibility and accuracy one must clearly ask *several* volunteers to assess each drug image and then form a consensus measurement which can be compared to the reference measurement. How should we form that consensus?

## Consensus

There are a range of methods one can use to extract a consensus from a set of classifications; the simplest being majority voting, however, this is not practical since an outright majority is not guaranteed. Alternatively one may take the mode, mean or median of the classifications, although the former

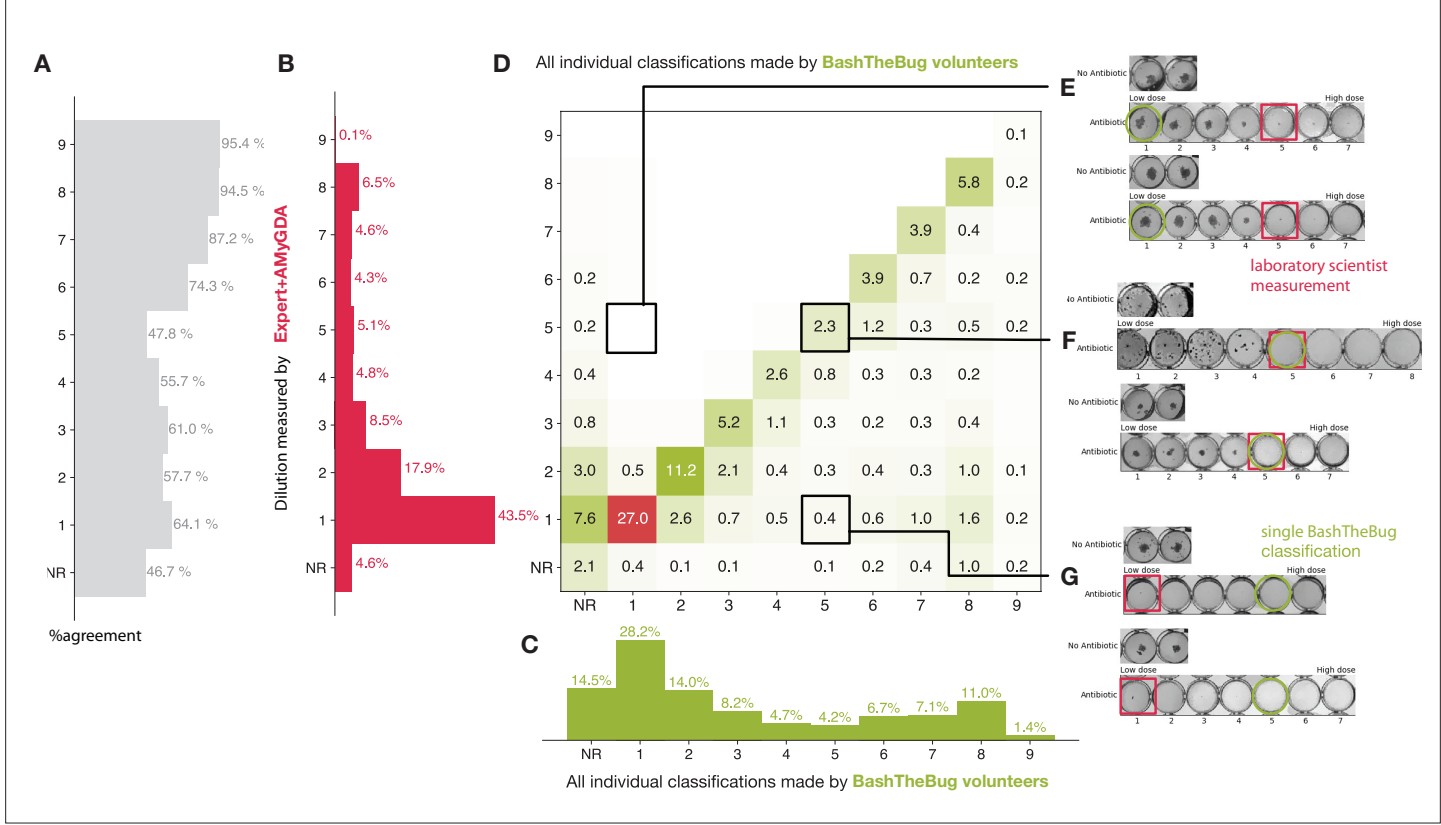

**Figure 2.** Heatmap showing how all the individual BashTheBug classifications (n=214,164) compare to the dilution measured by the laboratory scientist using the Thermo Fisher Vizion instrument after 14 days incubation (n=12,488). (**A**) The probability that a single volunteer exactly agrees with the Expert +AMyGDA dataset varies with the dilution. (**B**) The distribution of all dilutions in the Expert +AMyGDA dataset after 14 days incubation. The differences are due to different drugs having different numbers of wells as well as the varying levels of resistance in the supplied strains. NR includes both plates that could not be read due to issues with the control wells and problems with individual drugs such as skip wells. (**C**) The distribution of all dilutions measured by the BashTheBug volunteers. (**D**) A heatmap showing the concordance between the Expert +AMyGDA dataset and the classifications made by individual BashTheBug volunteers. Only cells with $gt_{0.1}$% are labelled. (**E**) Two example drug images where both the Expert and AMyGDA assessed the MIC as being a dilution of 5 whilst a single volunteer decided no growth could be seen in the image. (**F**) Two example drug images where both the laboratory scientist and a volunteer agreed that the MIC was a dilution of 5. (**G**) Two example drug images where the laboratory scientist decided there was no growth in any of the wells, whilst a single volunteer decided there was growth in the first four wells.

The online version of this article includes the following figure supplement(s) for figure 2:

**Figure supplement 1.** Heatmap showing how all the individual BashTheBug classifications (n=214,164) compare to the set of dilutions where the measurement made by the laboratory scientist using the Thermo Fisher Vizion instrument and a mirrored box after 14 days incubation concur (n=9402) (**A**).

is not always defined and the last two do not always yield an integer. More sophisticated methods, such as the weighted-majority algorithm (*Littlestone and Warmuth, 1989*), give weights to the classifiers based on their accuracy; however, this requires each volunteer to first classify a ground-truth dataset, which was not available at the start of the project. Given the high level of inequality in participation (*Figure 1C*), such methods would be very difficult to apply in practice in our case. We shall therefore limit ourselves here to considering only the mean, median and mode. Since these methods all require the classifications to be numerical, we excluded all readings where the Expert +AMyGDA measurement and/or half or over of the volunteers decided the drug image could not be read. If the classification distribution was bi-modal, then the lower value of the dilution is returned. If necessary, the mean or median were also rounded down.

## Reproducibility

To create two consensus measurements by the volunteers of each drug image, two separate sets of 17 classifications were drawn with replacement. By applying the relevant method (mean, median or

mode) a consensus dilution was arrived at for each set and then the two results compared. To begin with only drug images with 17 or more classifications were considered and this bootstrapping process was repeated ten times for each drug image in the Expert+AMyGDA dataset. Considering only those drug images taken after 14 days incubation (*Figure 3A*, *Supplementary file 1e*), they are more likely to exactly agree with one another when the mode was applied (89.2 ± 0.1%) than the median (86.9 ± 0.1%) or mean (74.3 ± 0.1%). For the essential agreement, we find that the mean now performs best (97.2 ± 0.1%), followed by the median (94.2 ± 0.1%) and mode (94.1 ± 0.1%). Here and throughout, the standard error of the mean is quoted. As one might expect, the reproducibility for MICs measured by two laboratory scientists, as quantified by essential agreement, was previously shown to be slightly higher, at 97.9 % (*Rancoita et al., 2018*).

Hence only the mean exceeds the threshold for reproducibility (*International Standards Organization, 2007*) when 17 classifications are used to build a consensus. Repeating the analysis for the drug images in the larger Expert dataset yields the same conclusion (*Figure 3—figure supplement 1*). The heatmaps (*Figure 3B*) show how two consensus measurements arrived at via the mean tend to be similar but not necessarily identical to one another, whilst two consensus measurements derived using the mode are more likely to agree with one another but also are more likely to arrive at very different values. The median sits in between these two extremes.

## Accuracy

Comparing the consensus measurements from the volunteers to the set of MICs in the Expert+AMyGDA dataset yields a different picture (*Figure 3C*). The mode exactly agrees with the reference 80.9 ± 0.1% of the time, followed by the median (78.1 ± 0.1%) and then mean (68.4 ± 0.1%). The mean, despite performing best for reproducibility, has the lowest level of essential agreement (as well as exact agreement) with the Expert+AMyGDA readings (88.5 ± 0.1%), with the median (90.2 ± 0.1%) and mode (91.0 ± 0.1%) both exceeding the 90% accuracy threshold (*International Standards Organization, 2007*).

The heatmaps show how the consensus dilution of the classifications made by the volunteers is much more likely to be higher than the Expert+AMyGDA measurement than lower (*Figure 3D*), regardless of the consensus method, indicating perhaps that volunteers are more likely than laboratory scientists to classify any dark regions in a well as bacterial growth, or that laboratory scientists are more willing to discount some features as artefacts for example air bubbles or sediment. Repeating the analysis using the Expert dataset (*Figure 3—figure supplement 1*) leads to lower values for the exact and essential agreements for all consensus methods – this is to be expected since this the Expert reference dataset contains a larger proportion of errors than the Expert+AMyGDA dataset.

## Which method to choose?

Despite being the most reproducible method as measured by essential agreement, we discount the mean since it suffers from relatively poor levels of exact agreement for both reproducibility and accuracy and its performance falls faster than the other methods when $n$ is decreased. The median and mode have very similar reproducibilities and accuracy and we conclude they perform equally well. We can infer from this that bi-modal classification distributions are rare and that the median is often identical to the mode.

## Reducing the number of classifications

Clearly it would be desirable and ethical to only require the volunteers to complete the minimum number of classifications to achieve an acceptable result. The simplest way to do this is to decrease the number of classifications, $n$, before a drug image is retired – this reduces both the reproducibility and accuracy of the consensus measurements (*Figure 4*, *Supplementary file 1e, f*), however perhaps not by as much as one might expect. The mean exceeds the essential agreement ≥ 95% reproducibility threshold for $n \geq 13$, whilst the mode and the median satisfy the accuracy criterion of essential agreement ≥ 90% for $n \geq 3$ and $n \geq 11$, respectively (*Figure 4*, *Supplementary file 1f*). Similar trends are observed when the Expert dataset is used as the reference (*Figure 4—figure supplement 1*) or when the plates are incubated for different periods of time (*Figure 4—figure supplements 2 and 3*). Accuracy is hence less sensitive than reproducibility to reducing the number of classifications used to

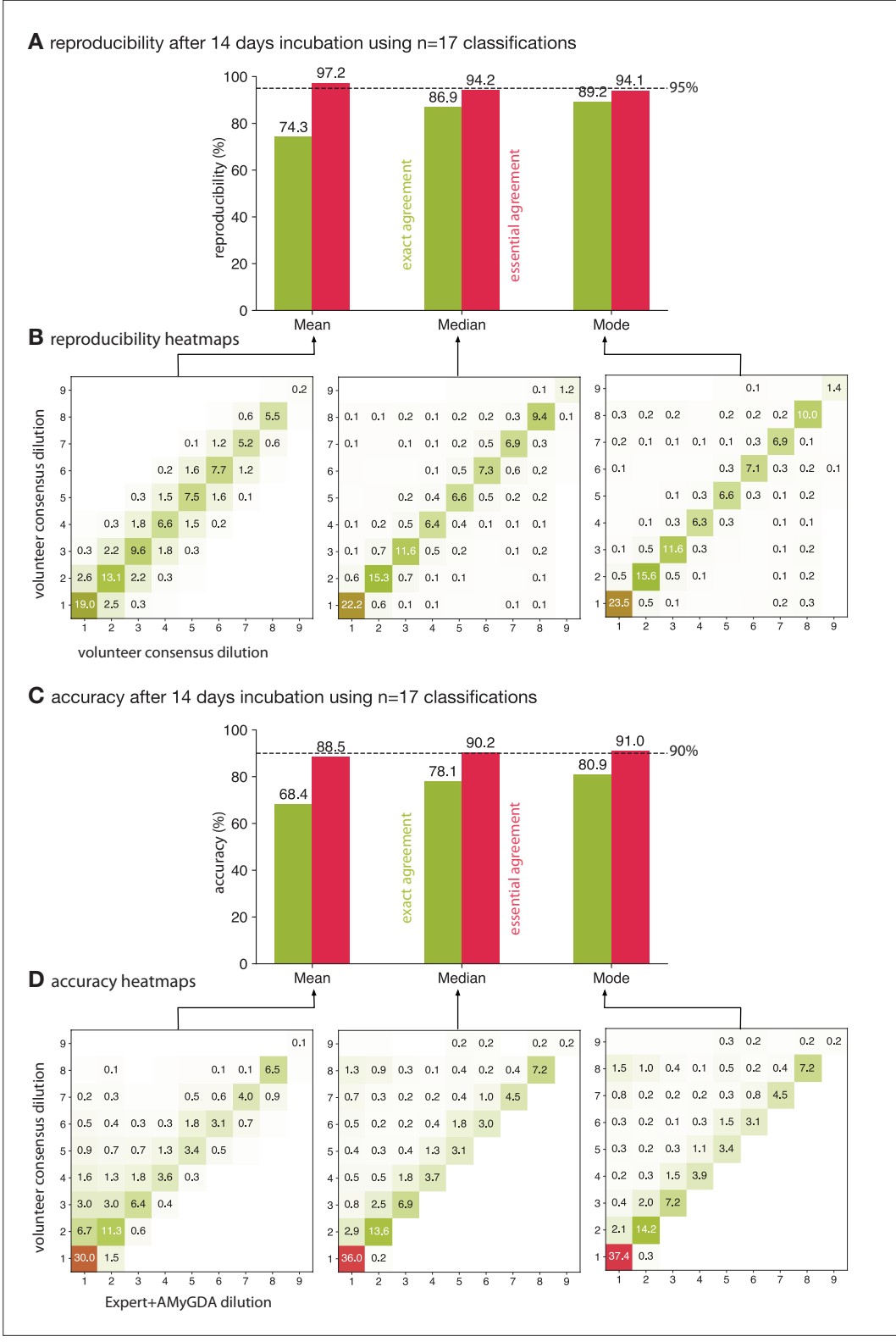

**Figure 3.** Taking the mean of 17 classifications is ≥95% reproducible whilst applying either the median or mode is ≥90% accurate. (**A**) Only calculating the mean of 17 classifications achieves an essential agreement ≥95% for reproducibility *International Standards Organization, 2007*, followed by the median and the mode. (**B**) Heatmaps of the consensus formed via the mean, median or mode after 14 days incubation. Only drug images from the Expert + AMyGDA dataset are included. (**C**) The essential agreement between a consensus dilution

*Figure 3 continued on next page*

*Figure 3 continued*

formed from 17 classifications using the median or mode and the consensus Expert +AMyGDA dilution both exceed the required 90% threshold *International Standards Organization, 2007*. (**D**) The heatmaps clearly show how the volunteer consensus dilution is likely to be the same or greater than the Expert + AMyGDA consensus.

The online version of this article includes the following figure supplement(s) for figure 3:

**Figure supplement 1.** Taking the mean of 17 classifications is ≥95% reproducible whilst none of the methods reach have an essential agreement for accuracy of ≥90% when using the Expert dataset.

build the consensus and, depending on the consensus method used, the number of classifications can be reduced whilst still maintaining acceptable levels of accuracy.

## Can we improve matters?

Retiring all drug images after a fixed number of classifications is simple but does not take account of the relative difficulty of the classification task. If one was able to group the drug images by difficulty, either before upload or dynamically during classification, then one could optimally target the work undertaken by the volunteers. Due to the inherent difficulties in culturing *M. tuberculosis*, there is a

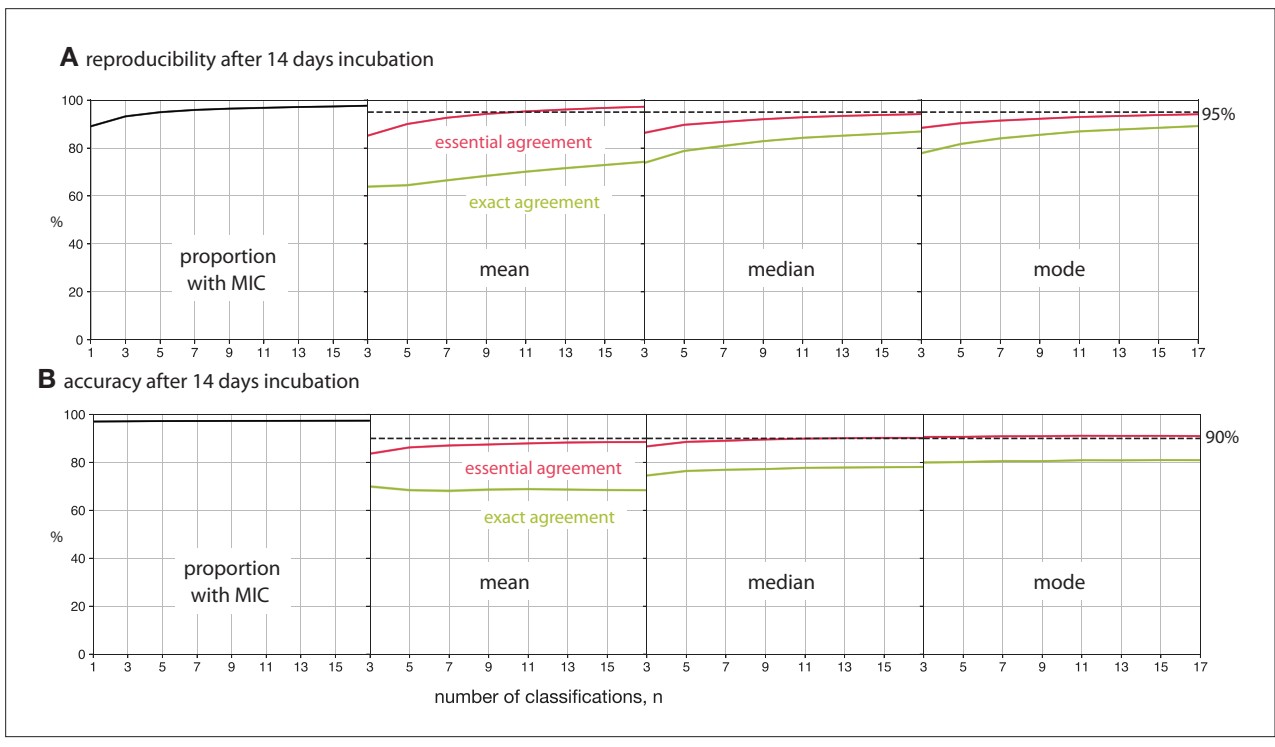

**Figure 4.** Reducing the number of classifications, *n*, used to build the consensus dilution decreases the reproducibility and accuracy of the consensus measurement. (**A**) The consensus dilution becomes less reproducible as the number of classifications is reduced, as measured by both the exact and essential agreements. (**B**) Likewise, the consensus dilution becomes less accurate as the number of classifications is decreased, however the highest level of exact agreement using the mean is obtained when *n* = 3 and the mode, and to a lesser extent the median, are relatively insensitive to the number of classifications. These data are all with respect to the Expert +AMyGDA dataset.

The online version of this article includes the following figure supplement(s) for figure 4:

**Figure supplement 1.** Reducing the number of classifications, *n*, used to build the consensus dilution decreases the reproducibility and accuracy of the consensus measurement.

**Figure supplement 2.** Altering the number of days incubation does not markedly affect the observed trends in reproducibility.

**Figure supplement 3.** Altering the number of days incubation does not markedly affect the observed trends in accuracy.

**Figure supplement 4.** Segmenting the drug images by the mean amount of growth in the positive control wells (*Figure 6—figure supplement 3*) does not markedly affect the reproducibility of the three consensus methods.

**Figure supplement 5.** Segmenting the drug images by the mean amount of growth in the positive control wells (*Figure 6—figure supplement 3*) does not markedly affect the accuracy of the three consensus methods.

broad distribution of growth in the positive control wells after 14 days incubation (*Figure 6—figure supplement 3*). One might expect that drug images with poor growth would be more challenging to classify, however, segmenting by low, medium and high growth shows the amount of growth in the positive control wells has little effect on either the reproducibility (*Figure 4—figure supplement 4*) or accuracy (*Figure 4—figure supplement 5*), regardless of the consensus method and number of classifications employed.

Alternatively one could use the first few classifications performed by the volunteers to assess the difficulty of each drug image. For example, if the first $n$ volunteers all return the same classification, then it is reasonable to assume that this is a straightforward image and it can be retired, with the remainder accruing additional classifications. Ideally one would want to develop an algorithm that assessed the likelihood of the classification not being altered by more classifications to allow a dynamic decision about when to halt, however, applying such an approach was not possible at the time in the Zooniverse.

To estimate the potential value in applying a simple approach to dynamically retiring drug images, we shall consider applying the median after 14 days of incubation and will arbitrarily retire a drug image if the first three volunteers all made the same classification, with all other drug images being retired after 17 classifications. This simple protocol reduces the number of classifications required to $n = 8.8$, a reduction of 48%, and the reproducibility, as measured by exact agreement, rises from 86.8% to 87.6%, whilst the essential agreement remains unchanged (94.2–94.4%). The accuracy, assessed in the same way, behaves similarly with the exact agreement increasing from 78.1% to 78.8% with the essential agreement remaining unaltered (90.2–90.3%). Hence retiring some of the drug images at $n = 3$ not only dramatically reduces the number of classifications required but also improves the result in a few cases, presumably because the subsequent classifications have a small chance of altering

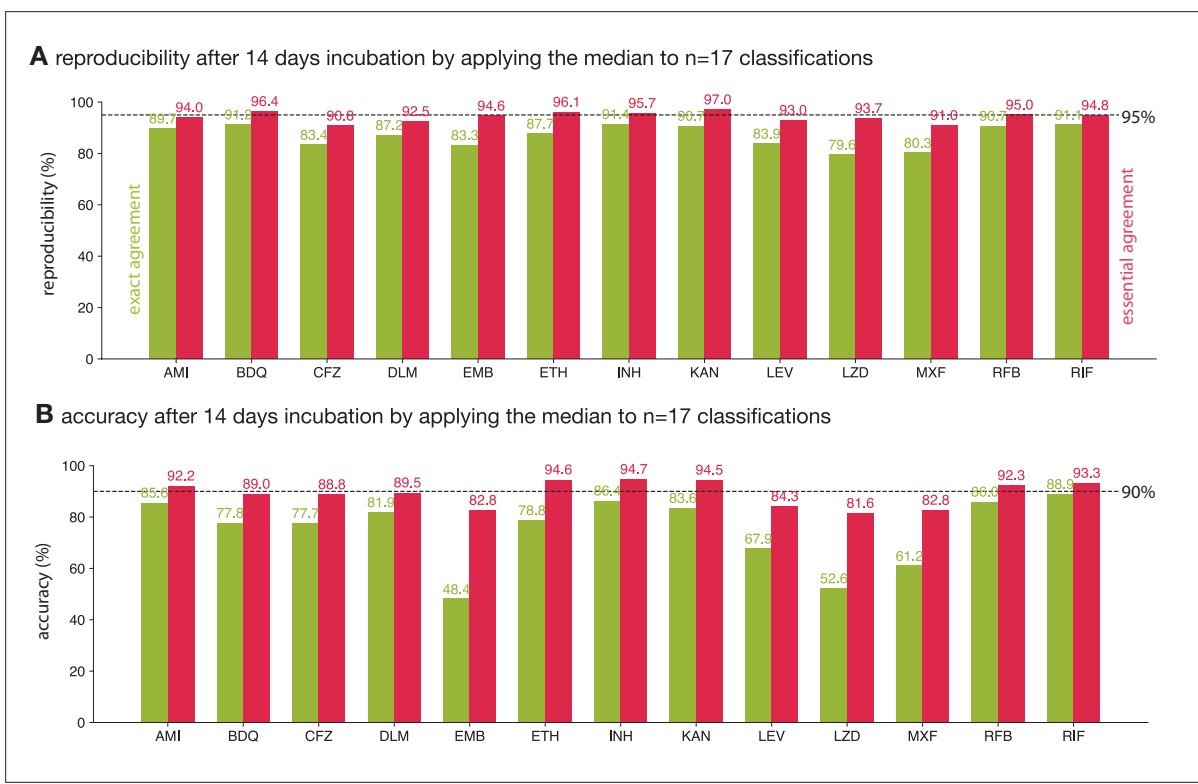

**Figure 5.** The reproducibility and accuracy of the consensus MICs varies by drug. Consensus MICs were arrived at by taking the median of 17 classifications after 14 days incubation. The essential and exact agreements are drawn as red and green bars, respectively. For the former the minimum thresholds required are 95% and 90% for the reproducibility and accuracy, respectively (*International Standards Organization, 2007*). See (*Figure 5—figure supplement 1*) for the other consensus methods.

The online version of this article includes the following figure supplement(s) for figure 5:

**Figure supplement 1.** The reproducibility and accuracy after 14 days incubation of the 13 antibiotics on the UKMYC5 plate.

the dilution by a single unit, hence worsening the exact agreement but not affecting the essential agreement.

A fairer test is to ask if this dynamic approach improves performance if we are constrained to a fixed number of total classifications: if we choose $n = 9$, then the reproducibility of the median (as measured by exact and essential agreements) improves from 83.0–92.2% to 87.6–94.4% and the accuracy, measured in the same way, improves slightly from 77.2–89.6% to 78.8–90.3%. We therefore conclude that even a simple dynamic approach to retiring images would minimise the work done by the volunteers / allow more images to be classified.

## Variation by drug

So far we have analysed the reproducibility and accuracy of consensus MICs obtained from a crowd of volunteers, thereby aggregating the results for each of the 13 anti-tuberculars (excl. PAS) present on the UKMYC5 plate design. The reproducibility of each drug, as measured by the exact and essential agreements, varies between 79.6–91.4% and 90.8–97.0%, respectively (*Figure 5A*). This is consistent with an earlier study which found that intra- and inter-laboratory reproducibility when the drugs were read by a scientist using a Vizion instrument varied between 93.4-99.3 % and 87.1-99.1 % (as measured by essential agreement, excl. PAS), respectively (*Rancoita et al., 2018*). Earlier we showed that the reproducibility of the whole plate under these conditions when assessed using the essential agreement is 94.2 ± 0.1% (*Figure 3A*) – this is below the 95% threshold specified by an international standard for aerobic bacteria (*International Standards Organization, 2007*). Applying the same threshold to each drug we find that five out of the 13 drugs meet or exceed the threshold whilst the plate as a whole does not. The accuracy of each drug varies more widely: between 48.4–88.9% and 82.8–94.7% when assessed using the exact and essential agreement, respectively. Hence whilst the accuracy of the plate as a whole was 90.2 ± 0.1%, just exceeding the 90% threshold, only six out of 13 drugs surpassed the same threshold.

The variation in reproducibility and accuracy between anti-tuberculars, as well as between the exact and essential agreement for a single compound, is due to a number of factors, not limited to the number and concentration range of the wells on the plate design, the mechanisms of both action and resistance, the prevalence of resistance and the degradation rate after lyophilisation, both during storage and after inoculation. For example, kanamycin passes the reproducibility and accuracy thresholds we have adopted and this is likely due to there being relatively few (five) drug wells on the UKMYC5 plate (*Figure 6—figure supplement 1*) and the mechanism of resistance being such a substantial proportion of samples either do not grow in any well, or grow in all the drug wells, making measurement more straightforward. The mechanism of action of each compound is likely to affect how 'easy' it is to determine the minimum inhibitory concentration. From the striking differences in exact and essential accuracies of reading ethambutol, moxifloxacin and linezolid we hypothesise the Mycobacterial growth diminishes more gradually with increasing concentration for these drugs, rather than coming to an abrupt end, as it does for other compounds.

This whole analysis could be considered somewhat moot since the UKMYC5 96-well plate would be treated as a single entity (or medical device) if accreditation were to be sought and therefore the results for individual compounds would likely not be considered. One unintended consequence of the current standards is therefore that one could improve the performance of a plate design by dropping compounds with lower-than-average performance, even if this is clinically not desirable, rather than work to for example improve the performance of the measurement methods.

## Discussion

A crowd of volunteers can reproducibly and accurately measure the growth of a bacterial pathogen on a 96-well broth microdilution plate, thereby demonstrating the potential for clinical microbiology to embrace and combine contrasting measurement methods. No Mycobacterial antibiotic susceptibility testing standard exists, although efforts are underway to establish a reference method (*Schön et al., 2019*), and so we applied the standard for antibiotic susceptibility testing of aerobic bacteria (*International Standards Organization, 2007*). Forming a consensus by applying the mode or median to 17 independent classifications performs better overall than the mean, and the reproducibility of both these methods, as measured by the essential agreement, is 94.2% and 94.1%, respectively (*Figure 3*).

This is below than the 95% threshold set by ISO for aerobic bacteria and therefore the volunteers do not pass this standard. The accuracy of the crowd, as measured by the essential agreement, is 90.2% and 91.0% when the median and mode, respectively, are applied to produce a consensus measurement – these values are above the required 90% threshold (*International Standards Organization, 2007*), and therefore the volunteers are sufficiently accurate (but not quite reproducible enough) to be classified as an antibiotic susceptibility testing (AST) device.

The BashTheBug project has therefore been successful in its primary aim, i.e. the consensus readings made by the crowd of volunteers have been successfully used within the CRyPTIC project to reduce the measurement error in the dataset of 268,281 (13 drugs × 20,637 samples) minimum inhibitory concentrations. CRyPTIC has already significantly contributed to the first resistance catalogue published by the World Health Organization (*Walker et al., 2022*), carried out the largest genome-wide association study of *M. tuberculosis* (*The CRyPTIC Consortium, 2021*) and assessed the magnitude of MIC-shifts for the most common mutations that confer resistance (*The CRyPTIC Consortium, 2021*). The intention is to accelerate the shift from culture-based to genetics-based AST, which is already faster and cheaper than the culture-based alternatives for tuberculosis (*Pankhurst et al., 2016*) and this is well underway (*Walker et al., 2017*). Ultimately shifting to a genetics-based paradigm, where the susceptibility of a pathogen to an antibiotic is inferred from the genome of the pathogen, potentially offers a route to making pathogen diagnostics much more widely available. On a more personal note, it has been very fulfilling engaging with the public through the BashTheBug project and we urge other scientists to consider citizen science if it is appropriate for their research. It is our experience that, whilst the sheer volume of classifications a crowd can achieve can be staggering, it requires us, as researchers, to invest time and effort in regularly communicating with the volunteers which we mainly achieved through our blog, Twitter account, the Zooniverse talk boards and newsletter emails.

The volunteers are fast, taking on average 3.5 s per drug image, and therefore a single plate requires slightly less than 13 min of volunteer time to read if 17 classifications are amassed for each drug. Reducing the number of classifications before an image is retired reduces the reproducibility and accuracy, but not by as much as one might expect. A more nuanced approach would be to retire a drug image early if the first few classifications are identical, however it was not possible to define this type of dynamical rule in the Zooniverse portal when these data were collected. The level of participation by the volunteers was very unequal with a small cadre of volunteers doing very large numbers with ten volunteers doing, on average, over 10,000 classifications each which is more than many of the laboratory scientists who are considered the experts!

Compared to the measurements taken by the laboratory scientists, the consensus dilution arrived at by the volunteers tends to be higher, indicating a bias to overcall (*Figure 3—figure supplement 1*), which is supported by anecdotal observations of people classifying drug images at public engagement events where they often choose a higher dilution 'to be on the safe side'. As volunteers do more classifications they become more experienced and, on the whole, the magnitude of the bias reduces but does not disappear (*Figure 6—figure supplement 5*). By contrast, the AMyGDA computer software has been noted to have the opposite bias – that is is more likely to undercall compared to the expert (*Fowler et al., 2018c*). At present in the CRyPTIC project these biases are ignored, that is if the AMyGDA MIC and the consensus reached by BashTheBug agree it is assumed that is a correct measurement. A more nuanced approach that took into account these systematic biases would, we expect, reduce the levels of measurement error in the CRyPTIC MIC dataset yet further and slightly increase the number of MICs where two or more measurement methods agree with one another (*CRyPTIC Consortium, 2022*).

The reproducibility and accuracy of any method used to read a 96-well microtitre plate, whether that is laboratory scientists using a Thermo Fisher Vizion instrument or citizen scientists visually examining drug images, depends on a range of factors from the prevalence of drug-resistant samples in the dataset to which drugs are included in the plate design and the number and concentrations of their allotted wells. For the UKMYC5 plate design, both the Expert and either the AMyGDA or BashTheBug measurements are more likely to agree with one another at low dilutions where there is little or no *M. tuberculosis* growth in the drug wells (*Supplementary file 1d, i*), hence reducing the number of resistant samples would artificially 'improve' performance yet the standards do not specify the degree of resistance in any test dataset (*International Standards Organization, 2007*). The requirement to

have quality control strains that have definite growth in all the drug wells unintentionally mitigates against this risk. The dataset used here was based on 19 external quality assessment strains and therefore whilst it included some degree of resistance for all 13 drugs, there was only a single strain resistant to clofazimine, bedaquiline or delamanid and no strain was resistant to linezolid (*Rancoita et al., 2018*). For isoniazid and rifampicin, eight and seven of the 19 EQA strains, respectively were resistant and hence the prevalence of resistance for these drugs is much greater than would be expected to be encountered in most countries. Clearly studies including a much more diverse range of strains, for example clinical isolates, would be more definitive. Since the 13 antituberculars on the UKMYC5 plate (*Figure 6—figure supplement 1*) also all perform differently (*Figure 5*) different plate designs will perform differently, which is important as it is the plate that would be accredited, rather than the individual compounds.

Although the primary aim of this study was to assess whether the measurements produced by a crowd of volunteers are sufficiently reproducible and accurate to help reduce the measurement error in datasets containing large numbers of microtitre plates (as is being collected by the CRyPTIC project and others) the resulting dataset of classifications is ideally suited to train machine-learning models. This is increasingly recognised as an important use of citizen science (*Trouille et al., 2019*) and one could envisage training a light-weight machine-learning algorithm able to run on a mobile device which, by taking a photograph of a 96-well plate, could automatically read the minimum inhibitory concentrations. The best use of such a device would likely be to act as a double check for readings taken by the laboratory scientist. Alternatively, one could build a hybrid approach where e.g. small crowds of experts could examine plates used in a clinical microbiology service – these could be particularly difficult drug images or could be a random sample for quality assurance purposes. This type of hybrid approach would also help with training laboratory scientists which would help reduce the barrier to using 96-well microtitre plates for *M. tuberculosis* AST in clinical microbiology laboratories, especially in low- and middle-income countries. Finally, it is likely that each volunteer has their own individual bias and variability and constructing consensus methods (*Zhu et al., 2014*) that take these into account would likely further improve the performance of crowds of citizen scientists.

## Materials and methods
### Plate design
The CRyPTIC project has collected a large number (20,637) of clinical TB samples, each having the MICs of a panel of 13 antibiotics measured using a 96-well broth microdilution plate with the majority of samples also undergoing whole genome sequencing. Samples were collected by 14 laboratories based in 11 countries spread across five continents and included all the major lineages and were biased towards drug resistance (*CRyPTIC Consortium, 2022*, *The CRyPTIC Consortium, 2022*). Two plate designs were used, with the second (UKMYC6) being an evolution of the first (UKMYC5) plate design. In this paper we shall focus on data collected using the UKMYC5 plate design which is itself a variant of the MYCOTB 96-well microdilution plate manufactured by Thermo Fisher containing 14 anti-TB drugs. UKMYC5 includes two repurposed compounds (linezolid and clofazimine) and two new compounds (delamanid and bedaquiline). Since 96-well plates have 8 rows and 12 columns, fitting 14 drugs, alongside two positive control wells, onto the plate necessitated a complex design (*Figure 6—figure supplement 1*).

Each of the antibiotics on the UKMYC5 plate is present at 5–8 concentrations, each double the previous. The smallest concentration that prevents growth of *M. tuberculosis* after two weeks of incubation is the minimum inhibitory concentration (MIC). Since *M. tuberculosis* is notoriously difficult to culture and inspect, relying on a reading taken by a single expert (laboratory scientist) would likely have led to high levels of errors in the dataset.

### Image dataset
The images considered by the crowd of volunteers originated from an earlier study whose aim was to assess if the UKMYC5 plate was sufficiently reproducible and accurate for use as a high-throughput research MIC measurement device (*Rancoita et al., 2018*). Seven CRyPTIC laboratories were each sent thirty one vials containing one of 19 external quality assessment (EQA) *M. tuberculosis* strains, including the reference strain H37Rv ATCC 27294 (*Kubica et al., 1972*). The EQA strains varied in their

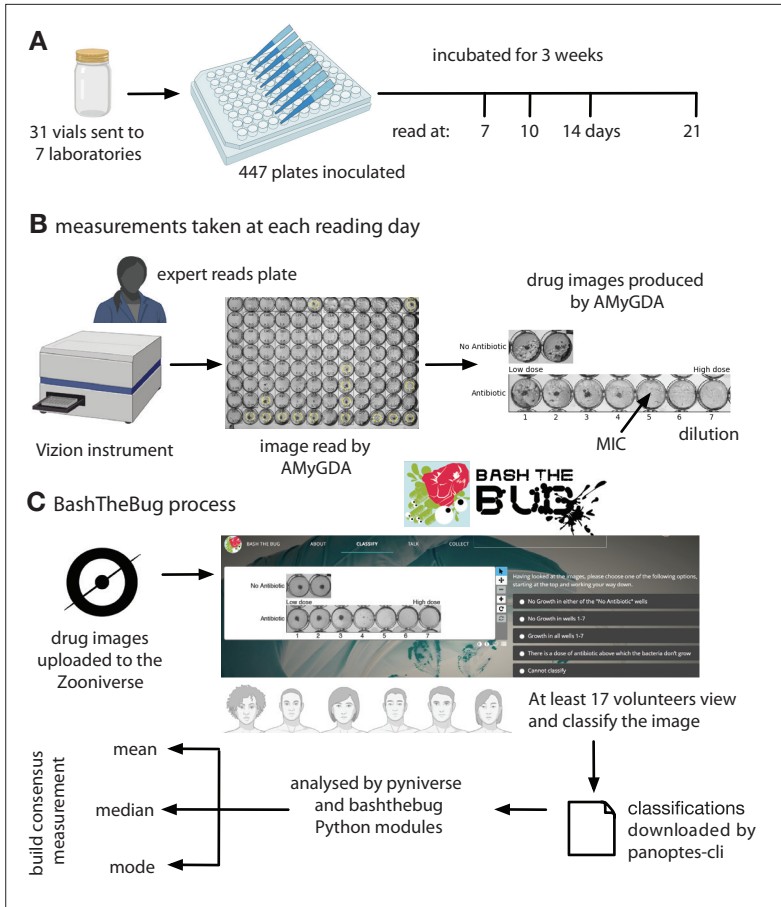

**Figure 6.** Each UKMYC5 plate was read by an Expert, by some software (AMyGDA) and by at least 17 citizen scientist volunteers via the BashTheBug project. (**A**) 447 UKMYC5 plates were prepared and read after 7, 10, 14 and 21 days incubation. (**B**) The minimum inhibitory concentrations (MIC) for the 14 drugs on each plate were read by an by Expert, using a Vizion instrument. The Vizion also took a photograph which was subsequently analysed by AMyGDA – this software then composited 14 drug images from each photograph, each containing an image of the two positive control wells. To allow data from different drugs to be aggregated, all MICs were converted to dilutions. (**C**) All drug images were then uploaded to the Zooniverse platform before being shown to volunteers through their web browser. Images were retired once they had been classified by 17 different volunteers. Classification data were downloaded and processed using two Python modules (pyniverse +bashthebug) before consensus measurements being built using different methods.

The online version of this article includes the following figure supplement(s) for figure 6:

**Figure supplement 1.** The UKMYC5 plate contains 14 different anti-TB drugs.

**Figure supplement 2.** Although the retirement limit within the Zooniverse platform was set to 17, over 1800 images received more classifications than this and a small number were only classified 15 or 16 times.

**Figure supplement 3.** The Expert +AMyGDA consensus dataset has the same distribution of bacterial growth in the positive control wells as the Expert dataset after 14 days incubation.

**Figure supplement 4.** The Expert +AMyGDA dataset has a greater proportion of drug images with low dilutions compared to the Expert dataset.

**Figure supplement 5.** The average bias per volunteer decreases with experience.

resistance profiles (*Rancoita et al., 2018*). Thirty of the vials were blinded and the last contained the reference strain. Within the thirty vials, some of EQA strains were duplicated and two of the blinded strains were also the reference strain. Since some labs only received a subset of the 31 vials (*Supplementary file 1a*), a total of 447 plates were inoculated and then incubated for 3 weeks (*Figure 6A*). Due to the slow growth rate of *M. tuberculosis*, minimum inhibitory concentrations of the 14 drugs on the plate were measured after 7, 10, 14, and 21 days by two scientists using a Thermo Fisher Sensititre

Vizion Digital MIC viewing system, a mirrored-box and a microscope. One or two photographs were also taken each time using the Vizion instrument (*Figure 6B*).

The study showed that the UKMYC5 plate is reproducible and that it is optimal to read the plate using either a Thermo Fisher Vizion instrument or a mirrored-box after 14 days of incubation (*Rancoita et al., 2018*). It also showed that *para*-aminosalicylic acid (PAS) was not reproducible on the plate and therefore this drug is excluded in all subsequent analyses. Each image was processed and analysed by some bespoke software, the Automated Mycobacterial Growth Detection Algorithm (AMyGDA), that segmented each photograph, thereby providing a second independent MIC reading of all the drugs on each plate (*Figure 6B*; *Fowler et al., 2018c*).

Early internal tests using the Zooniverse platform showed that asking a volunteer to examine all 96 wells on a plate was too arduous a task. We therefore observed a clinical microbiologist as she examined several photographs of UKMYC5 plates. Rather than considering the *absolute* growth in each well, she was constantly comparing the growth in the wells containing antibiotic back to the positive control wells and therefore was judging what constituted growth *relative* to how well the isolate had grown in the absence of drug. A suitable task is therefore to classify the growth in the wells for a single drug as long as the positive control wells are also provided. The AMyGDA software was therefore modified to composite such *drug images* (*Figure 6B*).

Each UKMYC5 plate yielded 14 composite images, one for each drug. Throughout the following analysis we shall aggregate all the data from the different drugs on the UKMYC5 plate. To facilitate this we shall therefore consider the *dilution*, which is defined as the number of the well in the drug image with the lowest antibiotic concentration which prevents bacterial growth, rather than the minimum inhibitory concentration. Following upload to the Zooniverse platform, the retirement threshold was set to 17 classifications, however some images attracted additional classifications with 421 images having ≥ 34 classifications whilst 89 have 100 or greater (*Supplementary file 1b*, *Figure 6—figure supplement 2*). This is due, we believe, to images not being removed when retired, but instead having a "Finished!" sticker added and hence images can accrue additional classifications, especially at times when the volunteers have completed all the uploaded images. Each volunteer is only permitted to classify each image once.

## Analysis

The resulting classifications were downloaded from the Zooniverse platform, either by a web browser or using the panoptes-cli command line tool (*McMaster et al., 2021*). Two Python modules were written to parse, store, manipulate and graph this classification data. The first, pyniverse (*Fowler, 2018a*), is designed to be generic for Zooniverse projects whilst the second, bashthebug (*Fowler, 2018b*), inherits from the first and adds functionality specific to BashTheBug (*Figure 6C*). Both are freely available to download and use. These Python modules output several Pandas (*McKinney, 2010*) dataframes which were then indexed, filtered and joined to other dataframes containing the sample information and the MIC readings taken by the expert and AMyGDA software. AMyGDA also measured the growth in the two positive control wells and this was also recorded in a dataframe. All subsequent analysis was performed using Python3 in a jupyter-notebook (see Data Availability Statement) and all graphs were plotted using matplotlib.

## Engagement

In addition to the Zooniverse project page, which contained background information, a tutorial and FAQs, we setup a blog (*Fowler, 2017*) and a variety of social media channels, focussing mainly on Twitter (@bashthebug). These all used a professionally designed logo and typeface (*Figure 6C*), allowing instantaneous recognition of the project, which is important since the Zooniverse platform hosts tens of projects at any one time, and to indirectly convey that this is a professional project and therefore of scientific and societal importance. Since the blog was launched in March 2017 we have written 71 posts, attracting 7,393 visitors who made 13,811 views. At the time of writing, the Twitter account, @bashthebug, has 393 followers and has tweeted 400 times. Finally, the volunteers interacted with one another as well as the project team via the BashTheBug Talk Boards on the Zooniverse platform. A total of 6,255 posts were made by 1042 individuals on 4834 topics. During the course of the project, one of our more experienced volunteers (EMLB) became a de facto moderator by

answering so many of the questions posted (>500) which we recognised by giving her moderator status and she is also an author of this paper.

## Acknowledgements

We are very grateful to the Zooniverse volunteer community (Figure 1—figure supplement 1) who contributed their time and energy to this project and to the Zooniverse development team for coding and maintaining the Zooniverse online platform. We thank David Hawkins for designing the BashTheBug logo and typeface and Chris Wood, Oxford Medical Illustration and Dr Nicola Fawcett, livinginamicrobialworld.com for the wild garden of the gut bacteria photographs. For the purpose of open access, the author has applied a CC BY public copyright licence to any Author Accepted Manuscript version arising from this submission. The views expressed are those of the author(s) and not necessarily those of the NHS, the NIHR or the Department of Health.

## Additional information

### Group author details

**The CRyPTIC Consortium**

**Ivan Barilar**: Research Center Borstel, Borstel, Germany; **Simone Battaglia**: IRCCS San Raffaele Scientific Institute, Milan, Italy; **Emanuele Borroni**: IRCCS San Raffaele Scientific Institute, Milan, Italy; **Angela Pires Brandao**: Oswaldo Cruz Foundation, Rio de Janeiro, Brazil; Institute Adolfo Lutz, Sao Paulo, Brazil; **Alice Brankin**: University of Oxford, Oxford, United Kingdom; **Andrea Maurizio Cabibbe**: IRCCS San Raffaele Scientific Institute, Milan, Italy; **Joshua Carter**: Stanford University School of Medicine, Stanford, United States; **Daniela Maria Cirillo**: IRCCS San Raffaele Scientific Institute, Milan, Italy; **Pauline Claxton**: Scottish Mycobacteria Reference Laboratory, Edinburgh, United Kingdom; **David A Clifton**: University of Oxford, Oxford, United Kingdom; **Ted Cohen**: Yale School of Public Health, Yale, United Kingdom; **Jorge Coronel**: Universidad Peruana Cayetano Heredia, Lima, Peru; **Derrick W Crook**: University of Oxford, Oxford, United Kingdom; **Sarah G Earle**: University of Oxford, Oxford, United Kingdom; **Vincent Escuyer**: Wadsworth Center, New York State Department of Health, Albany, United States; **Lucilaine Ferrazoli**: Institute Adolfo Lutz, Sao Paulo, Brazil; **Philip W Fowler**: University of Oxford, Oxford, United Kingdom; **George F Gao**: Chinese Center for Disease Control and Prevention, Beijing, China; **Jennifer Gardy**: Bill & Melinda Gates Foundation, Seattle, United States; **Saheer Gharbia**: UK Health Security Agency, London, United Kingdom; **Kelen Teixeira Ghisi**: Institute Adolfo Lutz, Sao Paulo, Brazil; **Arash Ghodousi**: IRCCS San Raffaele Scientific Institute, Milan, Italy; Vita-Salute San Raffaele University, Milan, Italy; **Ana Luıza Gibertoni Cruz**: University of Oxford, Oxford, United Kingdom; **Clara Grazian**: University of New South Wales, Sydney, Australia; **Jennifer L Guthrie**: The University of British Columbia, Vancouver, Canada; Public Health Ontario, Toronto, Canada; **Wencong He**: Chinese Center for Disease Control and Prevention, Beijing, China; **Harald Hoffmann**: SYNLAB Gauting, Munich, Germany; Institute of Microbiology and Laboratory Medicine,WHO-SRL Gauting, IMLred, Germany; **Sarah J Hoosdally**: University of Oxford, Oxford, United Kingdom; **Martin Hunt**: University of Oxford, Oxford, United Kingdom; EMBL-EBI, Hinxton, United Kingdom; **Zamin Iqbal**: EMBL-EBI, Hinxton, United Kingdom; **Nazir Ahmed Ismail**: National Institute for Communicable Diseases, Johannesburg, South Africa; **Lisa Jarrett**: UK Health Security Agency, Birmingham, United Kingdom; **Lavania Joseph**: National Institute for Communicable Diseases, Johannesburg, South Africa; **Ruwen Jou**: Taiwan Centers for Disease Control, Taipei, Taiwan; **Priti Kambli**: Hinduja Hospital, Mumbai, India; **Jeff Knaggs**: University of Oxford, Oxford, United Kingdom; EMBL-EBI, Hinxton, United Kingdom; **Anastasia Koch**: University of Cape Town, Cape Town, South Africa; **Donna Kohlerschmidt**: Wadsworth Center, New York State Department of Health, Albany, United States; **Samaneh Kouchaki**: University of Oxford, Oxford, United Kingdom; University of Surrey, Guildford, United Kingdom; **Alexander S Lachapelle**: University of Oxford, Oxford, United Kingdom; **Ajit Lalvani**: Imperial College, London, United Kingdom; **Simon Grandjean Lapierre**: Universite de Montreal, Canada, United States; **Ian F Laurenson**: Scottish Mycobacteria Reference Laboratory, Edinburgh, United Kingdom; **Brice Letcher**: EMBL-EBI, Hinxton, United Kingdom; **Wan-Hsuan Lin**: Taiwan Centers for Disease Control, Taipei,

Taiwan; **Chunfa Liu**: Chinese Center for Disease Control and Prevention, Beijing, China; **Dongxin Liu**: Chinese Center for Disease Control and Prevention, Beijing, China; **Kerri M Malone**: EMBL-EBI, Hinxton, United Kingdom; **Ayan Mandal**: The Foundation for Medical Research, Mumbai, India; **Daniela Matias**: UK Health Security Agency, Birmingham, United Kingdom; **Graeme Meintjes**: University of Cape Town, Cape Town, South Africa; **Flavia Freitas Mendes**: Institute Adolfo Lutz, Sao Paulo, Brazil; **Matthias Merker**: Research Center Borstel, Borstel, Germany; **Marina Mihalic**: Institute of Microbiology and Laboratory Medicine,WHO-SRL Gauting, IMLred, Germany; **James Millard**: Africa Health Research Institute, Durban, South Africa; **Paolo Miotto**: IRCCS San Raffaele Scientific Institute, Milan, Italy; **Nerges Mistry**: The Foundation for Medical Research, Mumbai, India; **David Moore**: Universidad Peruana Cayetano Heredia, Lima, Peru; London School of Hygiene and Tropical Medicine, London, United Kingdom; **Viola Dreyer**: Research Center Borstel, Borstel, Germany; **Darren Chetty**: Africa Health Research Institute, Durban, South Africa; **Kimberlee A Musser**: Wadsworth Center, New York State Department of Health, Albany, United States; **Dumisani Ngcamu**: National Institute for Communicable Diseases, Johannesburg, South Africa; **Hoang Ngoc Nhung**: Oxford University Clinical Research Unit, Ho Chi Minh City, Viet Nam; **Louis Grandjean**: University College London, London, United Kingdom; **Kayzad Soli Nilgiriwala**: The Foundation for Medical Research, Mumbai, India; **Camus Nimmo**: University College London, London, United Kingdom; **Nana Okozi**: National Institute for Communicable Diseases, Johannesburg, South Africa; **Rosangela Siqueira Oliveira**: Institute Adolfo Lutz, Sao Paulo, Brazil; **Shaheed Vally Omar**: National Institute for Communicable Diseases, Johannesburg, South Africa; **Nicholas Paton**: National University of Singapore, Singapore, Singapore; **Timothy EA Peto**: University of Oxford, Oxford, United Kingdom; **Juliana Maira Watanabe Pinhata**: Institute Adolfo Lutz, Sao Paulo, Brazil; **Sara Plesnik**: Institute of Microbiology and Laboratory Medicine,WHO-SRL Gauting, IMLred, Germany; **Zully M Puyen**: Instituto Nacional de Salud, Lima, Peru; **Marie Sylvianne Rabodoarivelo**: Institut Pasteur de Madagascar, Antananarivo, Madagascar; **Niaina Rakotosamimanana**: Institut Pasteur de Madagascar, Antananarivo, Madagascar; **Paola MV Rancoita**: Vita-Salute San Raffaele University, Milan, Italy; **Priti Rathod**: UK Health Security Agency, Birmingham, United Kingdom; **Esther Robinson**: UK Health Security Agency, Birmingham, United Kingdom; **Gillian Rodger**: University of Oxford, Oxford, United Kingdom; **Camilla Rodrigues**: Hinduja Hospital, Mumbai, India; **Timothy C Rodwell**: FIND, Geneva, Switzerland; University of California, San Diego, San Diego, United States; **Aysha Roohi**: University of Oxford, Oxford, United Kingdom; **David Santos-Lazaro**: Instituto Nacional de Salud, Lima, Peru; **Sanchi Shah**: The Foundation for Medical Research, Mumbai, India; **Thomas Andreas Kohl**: Research Center Borstel, Borstel, Germany; **Grace Smith**: UK Health Security Agency, London, United Kingdom; UK Health Security Agency, Birmingham, United Kingdom; **Walter Solano**: Universidad Peruana Cayetano Heredia, Lima, Peru; **Andrea Spitaleri**: IRCCS San Raffaele Scientific Institute, Milan, Italy; Vita-Salute San Raffaele University, Milan, Italy; **Philip Supply**: Institut Pasteur de Lille, Lille, France; **Adrie JC Steyn**: Africa Health Research Institute, Durban, South Africa; **Utkarsha Surve**: Hinduja Hospital, Mumbai, India; **Sabira Tahseen**: National TB Reference Laboratory, National TB Control Program, Islamabad, Pakistan; **Nguyen Thuy Thuong**: Oxford University Clinical Research Unit, Ho Chi Minh City, Viet Nam; **Guy Thwaites**: University of Oxford, Oxford, United Kingdom; Oxford University Clinical Research Unit, Ho Chi Minh City, Viet Nam; **Katharina Todt**: Institute of Microbiology and Laboratory Medicine,WHO-SRL Gauting, IMLred, Germany; **Alberto Trovato**: IRCCS San Raffaele Scientific Institute, Milan, Italy; **Christian Utpatel**: Research Center Borstel, Borstel, Germany; **Annelies Van Rie**: University of Antwerp, Antwerp, Belgium; **Srinivasan Vijay**: University of Edinburgh, Edinburgh, United Kingdom; **Timothy M Walker**: University of Oxford, Oxford, United Kingdom; University of Oxford, Oxford, United Kingdom; **A Sarah Walker**: University of Oxford, Oxford, United Kingdom; **Robin Warren**: Stellenbosch University, Cape Town, South Africa; **Jim Werngren**: Public Health Agency of Sweden, Solna, Sweden; **Ramona Groenheit**: Public Health Agency of Sweden, Solna, Sweden; **Maria Wijkander**: Public Health Agency of Sweden, Solna, Sweden; **Robert J Wilkinson**: Imperial College, London, United Kingdom; Wellcome Centre for Infectious Diseases Research in Africa, Cape Town, South Africa; Francis Crick Institute, London, United Kingdom; **Daniel J Wilson**: University of Oxford, Oxford, United Kingdom; **Penelope Wintringer**: EMBL-EBI, Hinxton, United Kingdom; **Yu-Xin Xiao**: Taiwan Centers for Disease Control, Taipei, Taiwan; **Yang Yang**: University of Oxford, Oxford, United Kingdom; **Zhao Yanlin**: Chinese Center for Disease Control and Prevention, Beijing, China; **Shen-Yuan Yao**: National Institute for

Communicable Diseases, Johannesburg, South Africa; **Baoli Zhu**: Institute of Microbiology, Chinese Academy of Sciences, Beijing, China; **Stefan Niemann**: Research Center Borstel, Borstel, Germany; German Center for Infection Research (DZIF), Hamburg-Lubeck-Borstel-Riems, Germany; **Max O'Donnell**: Columbia University Irving Medical Center, New York, United States

## Competing interests

The Zooniverse Volunteer Community: The CRyPTIC Consortium: The other authors declare that no competing interests exist.

## Funding

| Funder | Grant reference number | Author |
|---|---|---|
| Wellcome Trust | 200205/Z/15/Z | Philip W Fowler<br>Carla Wright<br>Sarah W Hoosdally<br>Ana L Gibertoni Cruz<br>Aysha Roohi<br>Samaneh Kouchaki<br>Timothy M Walker<br>Timothy EA Peto<br>David Clifton<br>Derrick W Crook<br>A Sarah Walker |
| Bill and Melinda Gates Foundation | OPP1133541 | Philip W Fowler<br>Carla Wright<br>Sarah W Hoosdally<br>Ana L Gibertoni Cruz<br>Aysha Roohi<br>Samaneh Kouchaki<br>Timothy M Walker<br>Timothy EA Peto<br>David Clifton<br>Derrick W Crook<br>A Sarah Walker |
| Wellcome Trust | 203141/Z/16/Z | Philip W Fowler |

The funders had no role in study design, data collection and interpretation, or the decision to submit the work for publication. For the purpose of Open Access, the authors have applied a CC BY public copyright license to any Author Accepted Manuscript version arising from this submission.

## Author contributions

Philip W Fowler, Conceptualization, Data curation, Formal analysis, Investigation, Methodology, Software, Visualization, Writing – original draft, Writing – review and editing; Carla Wright, Aysha Roohi, Funding acquisition, Writing – review and editing; Helen Spiers, Conceptualization, Investigation, Methodology, Writing – review and editing; Tingting Zhu, Samaneh Kouchaki, Formal analysis, Writing – review and editing; Elisabeth ML Baeten, Funding acquisition, Supervision; Sarah W Hoosdally, Methodology, Funding acquisition, Project administration, Writing – review and editing; Ana L Gibertoni Cruz, Conceptualization, Funding acquisition, Writing – review and editing; Timothy M Walker, Timothy EA Peto, Investigation, Writing – review and editing; Grant Miller, Conceptualization, Methodology, Software, Writing – review and editing; Chris Lintott, Conceptualization, Methodology, Writing – review and editing; David Clifton, Formal analysis, Writing - original draft, Writing – review and editing; Derrick W Crook, Writing - original draft, Writing – review and editing; A Sarah Walker, Conceptualization; The Zooniverse Volunteer Community, The CRyPTIC Consortium, Investigation;

## Author ORCIDs

Philip W Fowler (iD) http://orcid.org/0000-0003-0912-4483
Ana L Gibertoni Cruz (iD) http://orcid.org/0000-0002-9473-2215
Timothy M Walker (iD) http://orcid.org/0000-0003-0421-9264
Derrick W Crook (iD) http://orcid.org/0000-0002-0590-2850
A Sarah Walker (iD) http://orcid.org/0000-0002-0412-8509

Decision letter and Author response
Decision letter https://doi.org/10.7554/eLife.75046.sa1
Author response https://doi.org/10.7554/eLife.75046.sa2

## Additional files

### Supplementary files
• MDAR checklist

• Supplementary file 1. A supplementary file containing a tables (a-i) is available online. The majority of the tables in the supplemental file can also be reproduced using the accompanying jupyter notebook at https://github.com/fowler-lab/bashthebug-consensus-dataset; *Fowler Lab, 2022*.

### Data availability
The data tables and a Jupyter notebook that allows the user to recreate the majority of figures and tables in both the manuscript and the supplemental information is freely available here: https://github.com/fowler-lab/bashthebug-consensus-dataset, (copy archived at swh:1:rev:61ba253936f7e-c33a785158ee48c84e63256cd8b) .It is setup so a user can either clone the repository and run the jupyter-notebook on their local computer (the installation process having installed the pre-requisites) or by clicking the "Launch Binder" button in the README, they can access and run the jupyter-notebook via their web browser, thereby avoiding any installation. I've added a short statement to the manuscript -- please advise if you think it needs changing.

The following dataset was generated:

| Author(s) | Year | Dataset title | Dataset URL | Database and Identifier |
|---|---|---|---|---|
| Fowler PW | 2021 | BashTheBug dataset for finding the optimal consensus method | https://github.com/fowler-lab/bashthebug-consensus-dataset | GitHub, consensus-dataset |

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
