## [Editor Report]

The authors evaluate a novel crowd–sourcing method to interpret minimum inhibitory concentrations of *Mycobacterium tuberculosis*, the causative agent of tuberculosis. To provide valuable test results without the need for available expert mycobacteriologists, the authors demonstrate that when presented appropriately, 11–17 interpretations by lay interpreters can provide reproducible results for most tuberculosis drugs. This analysis demonstrates that among those samples that can be reliably interpreted by automated detection software, lay interpretation provides a potential alternative means to provide a timely confirmatory read. The work will be of interest to bacteriologists and those with an interest in antimicrobial resistance.

---

## [Decision Letter]

**Decision letter after peer review:**

Thank you for submitting your article "BashTheBug: a crowd of volunteers reproducibly and accurately measure the minimum inhibitory concentrations of 13 antitubercular drugs from photographs of 96–well broth microdilution plates" for consideration by *eLife*. Your article has been reviewed by 3 peer reviewers, and the evaluation has been overseen by Bavesh Kana as the Senior and Reviewing Editor. The reviewers have opted to remain anonymous.

Essential revisions:

Data presentation/interpretation/clarity:

1. While the authors explained how they try to engage people to "play" this serious game and describe the number of classifications, there is no real discussion about the number of users playing every day. Reaching a minimum number of regular players is essential for the sustainability of such project.

2. In the discussion, the authors mentioned that this approach may help training laboratory scientists unfortunately this claim was not really explored in this manuscript. It may have been interesting to analyze, for the most engaged volunteers, the improvement in term of accuracy of a user after 10, 100 or 1000 classifications. This may be also interesting to reweight the accuracy results in function of the users experience to see if it can improve the classification scores. It would have been also of interest to know if the way of presenting the data or playing the game may help experts (i.e. laboratory scientists) to improve their skills to quickly assess MIC using the methodology design to assess the citizens (such as time spent on a classification presented in Figure S5).

3. 13 drugs were tested on 19 different strains of Mtb. It would have been of broad interest to see then how to reconstruct each plate from the different classifications and briefly present the practical outputs of these classifications: i.e. the resistance of each strain to the different antibiotics. Furthermore, except H37rV, the other strains are not mentioned; only a vial code is presented in Table S1.

4. Explain why you decided on 17 classifications in particular? You seem to make several arguments for 9 or even 3 classifications, but the reason for 17 in particular is not clear. Why do some wells have more than 17 classifications: was this accident or design? If design, was it because of a lack of consensus about them, or some other reason?

Editorial:

1. In the abstract (lines 3–4), the authors refer to MIC testing as a conventional method, which is imprecise, as the majority of BSL3 labs that perform susceptibility testing for Mtb only assess a single critical concentration per drug. This is also a challenge for the Introduction, paragraph 3.

2. In the introduction p. 2 the authors mentioned: "(ii) to provide a large dataset of classifications to train machine–learning models". While, one can see how the datasets can be used for this purpose, there is no data in the current manuscript corroborating this claim. This sentence at this position in the introduction may need to be reformulate otherwise it can be seen as misleading for the reader looking for data about ML model training.

3. First paragraph: two very minor linguistic points, (1) Do you mean 1.4 million deaths worldwide in 2019, rather than restricted to a particular location, and this might provide additional context? (2) in the sentence "Ordinarily this is more than any other single pathogen, however SARS–CoV–2 killed more people than TB in 2020 and is likely to do so again in 2021", the flow of the paper would be clearer if the word "however" was replaced with "although" – it sounds as if the subject is about to change to the SARS–CoV–2 virus as it is.

4. Introduction, paragraph 1, line 3–4 – not all bacterial pathogens demonstrate increasing resistance, perhaps the term "most" would be more accurate

5. Are the datasets (Expert and Expert+AMyGDA) are available for reuse especially to see if other programs can compare with the current results?

6. p. 5 "however some images attracted additional classifications with 421 images having > 34 classifications". Can the authors comments why? Based on beginning of the sentence, one gets the sense that the impression that the images were retired after 17 classifications?

7. p. 8 Figure 1–A shows that the classifications were condensed in two periods and a long gap with no classifications from October 2017 to April 2019. Can the authors explain why? In the panel C, there is a bump at the duration equal to 25 s. Is there something happening in the program at that time?

8. p. 12 Figure 3, why is panel E is positioned in a blank square on panel D?

9. Methods, Image Dataset, paragraph 3, line 1: "asking a volunteers" should be changed to "asking volunteers" or "asking a volunteer"

10. Results, Time Spent, line 2: would rephrase, the "caused by e.g." section for improved readability.

11. Results, Expert Measurement, paragraph 3, line 2: the phrase "one candidate" is confusing, suggest "one possibility" or other phrasing.

12. Results, Variation by Drug: an additional consideration could be the role of drug position on the plate, as some drugs (such as delamanid) are known to degrade faster, and the central drugs of the plate appeared to be those with lower agreement. This could potentially be missed due to interpretation of higher than normal susceptibility to delamanid (as could be artificially elevated if the lyophilized drug on the plate were used >6 months from manufacture). That said, this is very hard to differentiate given that the entire plate, as discussed, is tested as a single unit, and the dataset represented limited resistance to clofazimine, bedaquiline, delamanid, and linezolid. Can you comment on this?

13. Are the authors willing to open the BashtheBug platform so that other labs – especially in developing countries – can provide datasets?

14. The mention that CRyPTIC is seeking to expand knowledge of genetic variants of TB and the reasons for this are very exciting. However, this sentiment is not echoed again in the manuscript. Some elaboration of this, in the context of your findings, might greatly strengthen your discussion, if you feel it is appropriate. For example, based on what the volunteers can do, what work do you plan to do based on it, have you learned anything new about the different genomes, and do you have any recommendations for other projects sequencing microbial genomes?

15. It would have been extremely interesting, if at all possible, to see what would have happened if several experts had classified the same plate – what might the errors have been then? You note that experts are inconsistent, but it might be interesting to calculate how much so, compared to volunteers. Can you comment on this?

16. Please explain why the intervals of 7, 10, 14 and 21 days of incubation were chosen for the analysis of the cultures? (The paper does not mention that TB cultures grow unusually slowly, and that of course this is an additional reason why treatment is difficult). Readers working in the field will doubtless be aware of this fact, but those in citizen science or other disciplines may not be.

17. Can you describe where the 20 000 cultures came from?

18. You describe, "Thirty one vials containing 19 external quality assessment (EQA) *M. tuberculosis* strains, including the reference strain H37Rv ATCC 272948", which "were sent to seven participating laboratories". How were these selected? Please add a summary of how this process took place. In the text, this paragraph is confusing because seven separate institutions were mentioned but only two members of staff – did these staff travel between institutions?

19. The paper would be clearer if there had been a separate section specifically on comparing Volunteer measurements with Expert and/or with Expert+AMyGDA.

20. Figure 1 (A, B and C) is too small.

21. It would be very useful to explain a little more fully exactly why "This is slightly less than the 95% threshold set by ISO for aerobic bacteria and therefore the volunteers do not need this criterion". The sentence implies the opposite, given that the volunteers do not meet the 95% threshold, this is exactly why they would need this criterion. Some clarification here might strengthen the main finding, that a crowd of volunteers can perform the required task.

22. It may possibly be worth quoting or referencing some literature on "the wisdom of crowds" or some such topic to explain more fully why you chose citizen science, and a large number of untrained eyes rather than a small number of trained ones, to analyse your dataset.

23. The authors mention several biases, e.g. the different error sources in the "expert" versus the "AMyGDA" analyses, and in the tendency of volunteers to pick high MICs while the AMyGDA picks low ones, but overall throughout the paper it is less clear how these were actually corrected for or otherwise how these bias were addressed.

*Reviewer #2 (Recommendations for the authors):*

Thank you for the opportunity to review your paper. I want to stress before I make my comments that I do lack expertise in the field of microbiology, and some of what I suggest might very justifiably be discounted on this basis. I cannot provide comments on your reagents or equipment, for example, but perhaps I can make some useful suggestions on paper clarity and the citizen science aspects.

Introduction:

First paragraph: two very minor linguistic points, (1) I presume you mean 1.4 million deaths worldwide in 2019, rather than restricted to a particular location, and this might provide additional context, (2) in the sentence "Ordinarily this is more than any other single pathogen, however SARS–CoV–2 killed more people than TB in 2020 and is likely to do so again in 2021", I feel the flow of the paper would be clearer if the word "however" was replaced with "although" – it sounds as if the subject is about to change to the SARS–CoV–2 virus as it is.

The mention that CRyPTIC is seeking to expand knowledge of genetic variants of TB and the reasons for this are very exciting. However, you don't mention them again. Some elaboration of this, in the context of your findings, might greatly strengthen your discussion, if you feel it is appropriate. For example, based on what the volunteers can do, what work do you plan to do based on it, have you learned anything new about the different genomes, and do you have any recommendations for other projects sequencing microbial genomes?

Methods:

I particularly appreciated the point about, having found that classifying 96 wells at a time was too arduous a task, the authors decided to watch what an expert did. It sounds as if the authors were willing to try out several methods before settling on one which was appropriate for volunteers. (Similarly, your sections describing "Engagement", for example, and from the looks of your social media, sound like you established a lively community, which is a great thing in citizen science.) This paragraph is not a recommendation to change anything – except to say that I hope you will do further citizen science projects in the future!

It would have been extremely interesting, if at all possible, to see what would have happened if several experts had classified the same plate – what might the errors have been then? You note that experts are inconsistent, but it might be interesting to calculate how much so, compared to volunteers. I do not suggest you redo any experiments; perhaps a short speculation or any findings you do have, even anecdotal, might suffice to provide insights here.

Please could an explanation be added of why you decided on 17 classifications in particular? You seem to make several arguments for 9 or even 3 classifications, and this comes into various discussions several times, but the reason for 17 in particular never became quite clear to me. I would also have been very interested to know why some wells had more than 17 classifications: was this accident or design? If design, was it because of a lack of consensus about them, or some other reason?

Please could an explanation also be added of why the intervals of 7, 10, 14 and 21 days of incubation were chosen for the analysis of the cultures? (The paper does not mention that TB cultures grow unusually slowly, and that of course this is an additional reason why treatment is difficult. Readers working in the field will doubtless be aware of this fact, but those in citizen science or other disciplines may not be, and I feel this paper will be valuable to those in other disciplines.) I noticed there were a great many references to classifications for 14 days, but very little seemed to be said about 7, 10 and 21 in comparison. I did not feel that Figure S12 really explained how the images themselves might change over time due to bacterial growth, or why 7, 10 and 21 days were paid comparatively little attention.

I would have felt reassured to know where CRyPTIC's 20,000 cultures came from. I presume they were taken from patients from a range of countries. Was patient consent required, or obtained, for their use in this project? I presume that procedures were followed, but I would recommend even a very brief clarification of this point.

Image Dataset:

I did not find it not clear whether the "Thirty one vials containing 19 external quality assessment (EQA) *M. tuberculosis* strains, including the reference strain H37Rv ATCC 272948", which "were sent to seven participating laboratories", were part of the 20,000–strong dataset or how exactly they were selected from this larger sample. The phrase which immediately follows this, "sent to seven participating laboratories as described previously", in fact refers to another paper, but I felt that the phrase "as described previously" hints that you describe it in this paper. I recommend that you add a summary of how this process took place, even a very short sentence. Additionally, I found this paragraph confusing because seven separate institutions were mentioned but only two members of staff – did these staff travel between institutions?

There is a clear section on "Expert Measurements", but I would have found the paper clearer if there had been a separate section specifically on comparing Volunteer measurements with Expert and/or with Expert+AMyGDA. (The section "Variability in classifications" certainly goes into this, but it felt to me that it brought together several issues at once, which made it less easy to understand.)

"The AMyGDA software was therefore modified to composite such drug images" is a sentence I do not quite understand, even when I look at Figure 1B. Does it mean that several images were stacked on top of each other? Or did it simply mean – as I generally understand – that the volunteers are not shown the entire image in one go, but rather they are shown a small section of it with a line of wells for each particular type of antibiotic plus the control?

I found Figure 1 (A, B and C) much too small to really be able to see what was happening clearly. On print, it would be very difficult to see! The later figures illustrated the methods and the Zooniverse interface far better. Figure 1 is very valuable and I recommend the authors consider expanding it, even if it takes up more space.

"How to compare?":

I had difficulty understanding the terms "exact agreement" and "essential agreement", which may simply be due to my own limited knowledge. However, I found myself doing a literature search for the terms and found nothing – which is understandable, given that you have created new key metrics. I do not quite understand the need for new key metrics – perhaps the authors might consider stating why established metrics were inadequate in this case. I found it difficult to find the reference provided in the same paragraph regarding the creation of new AST methods, as there were a great many with much the same name, the first one of which was a withdrawn paper (written in 2007 but revised in 2021), so a DOI in the reference list might be helpful here if you feel it is worthwhile.

Discussion:

It would be very useful to explain a little more fully exactly why "This is slightly less than the 95% threshold set by ISO for aerobic bacteria and therefore the volunteers do not need this criterion". The sentence implies the opposite, given that the volunteers do not meet the 95% threshold, this is exactly why they would need this criterion. Some clarification here might strengthen the main finding, that a crowd of volunteers can perform the required task.

I was very interested by the point that "… one could build a hybrid approach where e.g. small crowds of experts could examine plates used in a clinical microbiology service – these could be particularly difficult drug images or could be a random sample for quality assurance purposes." I recommend that random samples are always included in citizen science projects even when you want human volunteers to focus on difficult images (so that artificial intelligence can do the easier ones), because training volunteers only on difficult images means that they have no clear idea what a representative sample looks like and this could create a bias or other problems. There was an interesting case in the Zooniverse project Snapshot Serengeti, in which volunteers were asked to classify animals in photographs taken by camera traps. Some photographs only showed grass and no animals, and these images were removed because of the ethic of not wasting volunteer time. However, surprisingly, this actually decreased volunteer engagement, because essentially if every image is special then no image is special. (The Snapshot Serengeti team produced a poster about this, but to the best of my knowledge it was not written up in an academic paper – however, the Zooniverse team will probably be able to confirm that these events took place and provide you with a reference.) It is important that a human volunteer knows what a "weird" image is and how it is different from a standard one.

General points:

It may possibly be worth quoting or referencing some literature on "the wisdom of crowds" or some such topic to explain more fully why you chose citizen science, and a large number of untrained eyes rather than a small number of trained ones, to analyse your dataset.

The authors mention several biases, e.g. the different error sources in the "expert" versus the "AMyGDA" analyses, and in the tendency of volunteers to pick high MICs while the AMyGDA picks low ones, but overall throughout the paper I sometimes felt less clear on how these were actually corrected for or otherwise how these bias were addressed.

Overall, these are very minor criticisms of what to me was a very interesting paper. I wish you all the best with its publication, and for your future work building on these results.

---

## [Author Response]

Essential revisions:Data presentation/interpretation/clarity:1. While the authors explained how they try to engage people to "play" this serious game and describe the number of classifications, there is no real discussion about the number of users playing every day. Reaching a minimum number of regular players is essential for the sustainability of such project.

This is a good point. We have added a small panel to Figure 2 showing the distribution of number of active users per day and some text to the Volunteers subsection in the Results.

2. In the discussion, the authors mentioned that this approach may help training laboratory scientists unfortunately this claim was not really explored in this manuscript. It may have been interesting to analyze, for the most engaged volunteers, the improvement in term of accuracy of a user after 10, 100 or 1000 classifications. This may be also interesting to reweight the accuracy results in function of the users experience to see if it can improve the classification scores. It would have been also of interest to know if the way of presenting the data or playing the game may help experts (i.e. laboratory scientists) to improve their skills to quickly assess MIC using the methodology design to assess the citizens (such as time spent on a classification presented in Figure S5).

We have calculated the average bias for each volunteer (defined as the mean of all the differences between their dilution and the dilution of the Expert+AMyGDA dataset) and then plotted this against the total number of classifications each volunteer has made (Figure S17). This suggests that as volunteers get more experienced their bias reduces but does not disappear and we have added a sentence to the Discussion making this point and referring to the new figure. We did consider reweighting the scores but to have done this we would have needed to get all volunteers to classify a small set of known images to establish their weight which we would then prospectively apply to their classifications.

3. 13 drugs were tested on 19 different strains of Mtb. It would have been of broad interest to see then how to reconstruct each plate from the different classifications and briefly present the practical outputs of these classifications: i.e. the resistance of each strain to the different antibiotics. Furthermore, except H37rV, the other strains are not mentioned; only a vial code is presented in Table S1.

This is related to §2.16, §2.18. The resistance profile of the 19 EQA strains is described in detail in a previous study (https://doi.org/10.1128/AAC.00344-18) which we did not flag well enough in the manuscript. We’ve added a sentence pointing out that the EQA strains had different resistance profiles and cited the above reference.

4. Explain why you decided on 17 classifications in particular? You seem to make several arguments for 9 or even 3 classifications, but the reason for 17 in particular is not clear. Why do some wells have more than 17 classifications: was this accident or design? If design, was it because of a lack of consensus about them, or some other reason?

We wanted each image to be classified enough times so we could analyse the effect of number of classifications on reproducibility and accuracy but had to balance this against the ethics of asking our volunteers to do classifications that were essentially spurious i.e. their work was not changing our conclusions. An odd number was chosen to make finding the median easier.

Editorial:1. In the abstract (lines 3–4), the authors refer to MIC testing as a conventional method, which is imprecise, as the majority of BSL3 labs that perform susceptibility testing for Mtb only assess a single critical concentration per drug. This is also a challenge for the Introduction, paragraph 3.

The reviewers are correct, these are imprecise statements. We have reworded parts of the abstract and the introductions to correct this.

2. In the introduction p. 2 the authors mentioned: "(ii) to provide a large dataset of classifications to train machine–learning models". While, one can see how the datasets can be used for this purpose, there is no data in the current manuscript corroborating this claim. This sentence at this position in the introduction may need to be reformulate otherwise it can be seen as misleading for the reader looking for data about ML model training.

We agree this is potentially confusing and any discussion about using the dataset for machine-learning is best left to the Discussion so have removed this the mention of machine-learning from the Introduction.

3. First paragraph: two very minor linguistic points, (1) Do you mean 1.4 million deaths worldwide in 2019, rather than restricted to a particular location, and this might provide additional context? (2) in the sentence "Ordinarily this is more than any other single pathogen, however SARS–CoV–2 killed more people than TB in 2020 and is likely to do so again in 2021", the flow of the paper would be clearer if the word "however" was replaced with "although" – it sounds as if the subject is about to change to the SARS–CoV–2 virus as it is.

Yes, 1.4 million people were estimated to have died worldwide from TB in 2019. Agree with both points and have changed the text.

4. Introduction, paragraph 1, line 3–4 – not all bacterial pathogens demonstrate increasing resistance, perhaps the term "most" would be more accurate

Agreed, changed.

5. Are the datasets (Expert and Expert+AMyGDA) are available for reuse especially to see if other programs can compare with the current results?

Yes, the GitHub repository at https://github.com/fowler-lab/bashthebugconsensus-dataset contains the data tables (as CSVs) in the tables/ folder and there is a data schema provided. The PHENOTYPES table contains one row per drug per sample per reading day and has columns for each of the different reading methods (this is explained in the README.md file). For example, the Expert reading (given as a dilution) is recorded in the VZ column and the AMyGDA reading is recorded in the IM column. These are given as dilutions i.e. the MIC is the number of the well counting up from the lowest concentration starting at one. A negative number indicates that either the control wells on the plate failed or that this drug could not be read due to e.g. skip wells or contamination.

6. p. 5 "however some images attracted additional classifications with 421 images having > 34 classifications". Can the authors comments why? Based on beginning of the sentence, one gets the sense that the impression that the images were retired after 17 classifications?

We think this is because when an image reaches the retirement limit it is not removed from the pool of images shown to the volunteers but instead a “Finished!” sticker is shown on the top left part of the image. Hence it is possible for images to be classified more times than the retirement limit and this is likely to happen when all the current uploaded images have been retired. We have spoken to the Zooniverse team about this, and they are aware, but it is unclear to us if, in this situation, some images are more likely than others to be served to volunteers thereby exacerbating the problem, or whether it is purely random.

We have added a sentence explaining this to the text.

7. p. 8 Figure 1–A shows that the classifications were condensed in two periods and a long gap with no classifications from October 2017 to April 2019. Can the authors explain why? In the panel C, there is a bump at the duration equal to 25 s. Is there something happening in the program at that time?

During the first period the retirement limit was set to eleven (which is also the Zooniverse default value), however during early analysis of the data it became apparent that we would need additional classifications so we could analyse in more detail the effect of adding additional classifications, hence the second period. In the period in between the volunteers were working on images taken from real plates collected by the project.

We don’t know why there is a bump at 25 seconds! We suspect, but do not know for sure, that it is a rounding artefact. Since this duration is much greater than the modal duration of

3.5 seconds, we have not commented further on it in the manuscript.

8. p. 12 Figure 3, why is panel E is positioned in a blank square on panel D?

In Figure 3D each cell in the heatmap is annotated with the percentage of images if it is greater than 0.1%. Hence Figure 3E is pointing at a cell which has less than 0.1% of images and so appears blank but actually contains a number of classifications. We’ve made this clearer in the legend of Figure 3.

9. Methods, Image Dataset, paragraph 3, line 1: "asking a volunteers" should be changed to "asking volunteers" or "asking a volunteer"

Changed, thanks.

10. Results, Time Spent, line 2: would rephrase, the "caused by e.g." section for improved readability.

Thanks, have rephrased.

11. Results, Expert Measurement, paragraph 3, line 2: the phrase "one candidate" is confusing, suggest "one possibility" or other phrasing.

Have rephrased as suggested.

12. Results, Variation by Drug: an additional consideration could be the role of drug position on the plate, as some drugs (such as delamanid) are known to degrade faster, and the central drugs of the plate appeared to be those with lower agreement. This could potentially be missed due to interpretation of higher than normal susceptibility to delamanid (as could be artificially elevated if the lyophilized drug on the plate were used >6 months from manufacture). That said, this is very hard to differentiate given that the entire plate, as discussed, is tested as a single unit, and the dataset represented limited resistance to clofazimine, bedaquiline, delamanid, and linezolid. Can you comment on this?

This is a good question and was explicitly tackled by an earlier paper that sought to validate the UKMYC5 plate for use in research antimicrobial susceptibility testing (doi: 10.1128/AAC.00344-18) – we have therefore added a direct comparison to the results in that study and cited it.

13. Are the authors willing to open the BashtheBug platform so that other labs – especially in developing countries – can provide datasets?

Yes, but it depends on what you mean by open! At its heart BashTheBug is a project on the Zooniverse platform which has admin users who can modify tasks, upload images etc. If it wasn’t too much work, we could refactor the project to match their 96-well plate and upload the images – this should not be too hard since the software (AMyGDA) which segments the photographs of the 96-well plates is designed to accept different plate designs. If we didn’t have the time, the alternative would be to make them admin users and then they could upload their own images and retrieve the results, thereby making use of the BashTheBug brand and volunteer base.

14. The mention that CRyPTIC is seeking to expand knowledge of genetic variants of TB and the reasons for this are very exciting. However, this sentiment is not echoed again in the manuscript. Some elaboration of this, in the context of your findings, might greatly strengthen your discussion, if you feel it is appropriate. For example, based on what the volunteers can do, what work do you plan to do based on it, have you learned anything new about the different genomes, and do you have any recommendations for other projects sequencing microbial genomes?

We agree – the discussion about CRyPTIC in the Introduction felt misplaced as it left the reader “hanging” so we have moved it to the second paragraph of the Discussion and expanded upon it, including citing several of the early primary research outputs from the project as well as some personal thoughts on the experience, as researchers, of setting up and running a citizen science project.

15. It would have been extremely interesting, if at all possible, to see what would have happened if several experts had classified the same plate – what might the errors have been then? You note that experts are inconsistent, but it might be interesting to calculate how much so, compared to volunteers. Can you comment on this?

The CRyPTIC project did just this in a previous publication

(https://doi.org/10.1128/AAC.00344-18). The main aim of the paper was to validate the UKMYC5 plate design and determine the optimal reading methods for a laboratory scientist and also the optimal incubation time. Each plate was read by two scientists in each laboratory and hence the inter-reader agreement was measured as a function of incubation time (Figure 2B in the above reference). We have added a sentence to the Results comparing the reproducibility of the volunteers to this existing publication.

16. Please explain why the intervals of 7, 10, 14 and 21 days of incubation were chosen for the analysis of the cultures? (The paper does not mention that TB cultures grow unusually slowly, and that of course this is an additional reason why treatment is difficult). Readers working in the field will doubtless be aware of this fact, but those in citizen science or other disciplines may not be

The reviewer is correct that we’ve assumed a reader will know that TB cultures grow unusually slowly so have inserted a sentence to the Introduction to make this point. In addition (and this is related to §2.18) we have also rewritten the Image Dataset section in the Methods to make it clearer that the images came from the earlier study which was setup to determine the optimal incubation period, and it used 7, 10, 14 and 21 days.

17. Can you describe where the 20 000 cultures came from?

Fortunately, the main CRyPTIC analysis papers have been pre-printed (with a few in press) since this manuscript was submitted to *eLife* and we can therefore cite these to explain in more detail how the 20,637 samples were collected, which countries they originated from, the mix of lineages, the prevalence of resistance etc. We have rewritten the first paragraph of the Results, including citing two of the CRyPTIC studies to provide more context.

18. You describe, "Thirty one vials containing 19 external quality assessment (EQA) M. tuberculosis strains, including the reference strain H37Rv ATCC 272948", which "were sent to seven participating laboratories". How were these selected? Please add a summary of how this process took place. In the text, this paragraph is confusing because seven separate institutions were mentioned but only two members of staff – did these staff travel between institutions?

The EQA dataset was constructed early in the CRyPTIC project and hence the seven participating laboratories were simply the first to join the consortium. We were reluctant to spend too long describing how the dataset is constructed since that is described in detail in the citation but have added some text explaining a little more how the dataset was built and clarifying that the two members of staff were from each laboratory.

19. The paper would be clearer if there had been a separate section specifically on comparing Volunteer measurements with Expert and/or with Expert+AMyGDA.

We considered this when writing the manuscript but instead decided we should primarily consider the dataset that is likely to have fewer errors (Expert+AMyGDA) but acknowledge its possible problems and then after each analysis, refer to the Expert dataset as well to check that the conclusion still holds.

20. Figure 1 (A, B and C) is too small.

We have made it larger.

21. It would be very useful to explain a little more fully exactly why "This is slightly less than the 95% threshold set by ISO for aerobic bacteria and therefore the volunteers do not need this criterion". The sentence implies the opposite, given that the volunteers do not meet the 95% threshold, this is exactly why they would need this criterion. Some clarification here might strengthen the main finding, that a crowd of volunteers can perform the required task.

The volunteers did not meet the 95% threshold for reproducibility, and we are straightforwardly acknowledging that here. It is, however, a little nuanced since there is no ISO standard for Mycobacterial AST devices and so we are applying the standard for aerobic bacteria. We have reworded the sentence slightly.

22. It may possibly be worth quoting or referencing some literature on "the wisdom of crowds" or some such topic to explain more fully why you chose citizen science, and a large number of untrained eyes rather than a small number of trained ones, to analyse your dataset.

Thanks for drawing our attention to this; we have added a sentence in the concluding paragraph of the Introduction referring to Francis Galton’s Nature letter *Vox Populi* in 1907 which appears to be a plausible originator of this idea.

23. The authors mention several biases, e.g. the different error sources in the "expert" versus the "AMyGDA" analyses, and in the tendency of volunteers to pick high MICs while the AMyGDA picks low ones, but overall throughout the paper it is less clear how these were actually corrected for or otherwise how these bias were addressed.

We have broken out the section discussing biases into its own paragraph in the Discussion and have stated that CRyPTIC at present ignores these biases, i.e. it assumes that if MICs measured by AMyGDA and BashTheBug agree then the measurement is correct. We have also made the point that taking the biases into account would likely reduce the measurement error yet further as well as increasing the number of samples

where two or more measurement methods agree (since agree would be more nuanced than “has the same MIC”).